# Reasoning Quality Emerges Early: Data Curation for Reasoning Models

**Hongyi Henry Jin** [1]  **Wenhan Yang** [1]  **Meysam Ghaffari** [2]  **Carlos Morato** [2]  **Baharan Mirzasoleiman** [1]

## Abstract

Supervised fine-tuning (SFT) on a small, high-quality set of long reasoning traces is an effective approach for eliciting strong reasoning capabilities in Large Language Models (LLMs). However, existing methods for curating high-quality SFT data rely heavily on strong reasoning models to filter examples based on diversity and difficulty, making the curation process costly while often yielding suboptimal data quality. In this work, we show that diverse and challenging reasoning examples can be identified using only the initial reasoning tokens. Specifically, we demonstrate that difficult problems can be reliably detected based on the loss of the first 100 reasoning tokens evaluated at a randomly perturbed checkpoint of the pretrained model. We further show that examples exhibiting similar loss patterns over their first 1k reasoning tokens across a small number of perturbed checkpoints extrapolating along the fine-tuning trajectory provably induce similar gradients. We validate our approach through extensive experiments on fine-tuning Qwen2.5-7B and Llama3.1-8B models on the M23K medical reasoning and OpenThoughts-Math datasets. Our method outperforms existing baselines by up to 1.7% while being 91% more token efficient. [1]

## 1. Introduction

Recent large language models, such as DeepSeek-R1 (Guo et al., 2025) and o3 (OpenAI, 2024), exhibit strong performance on reasoning-intensive benchmarks spanning mathematics, programming, and scientific domains. These capabilities are induced through post-training a strong pretrained model with either reinforcement learning (RL) on very large data, or supervised fine-tuning (SFT) on a smaller set of examples with long reasoning traces. This encourages the model to generate explicit intermediate reasoning steps at inference time. The resulting long-form reasoning traces, often referred to as chains of thought (CoT) or "thinking tokens," have been shown to substantially improve solution accuracy (Javanmard et al., 2026b).

In contrast to RL, where scale often plays a more significant role than data quality, SFT is highly sensitive to the quality of the reasoning traces. Indeed, naively scaling SFT data without quality control can substantially degrade performance (Akter et al., 2025; Javanmard et al., 2026a). High-quality SFT datasets are commonly characterized by diversity and difficulty, i.e., examples that are challenging for the target model (Muennighoff et al., 2025; Guha et al., 2025). To meet these criteria, reasoning traces are either manually curated or generated by strong reasoning models, such as DeepSeek-R1, and subsequently filtered to promote diversity and difficulty (Muennighoff et al., 2025; Guha et al., 2025; Huang et al., 2025). Diversity is often enforced by categorizing data using more capable LLMs. Difficulty-based filtering prompts an LLM (e.g., GPT-4o-mini) to estimate the difficulty of each question and retains only the hardest instances, or selects questions that elicit the longest reasoning traces. However, the extensive LLM-based filtering is very expensive and often yields suboptimal results.

In this work, we show that, perhaps surprisingly, both the diversity and difficulty of reasoning traces can be inferred from the initial portion of the chain-of-thought (CoT). This observation enables early identification and efficient elimination of non-informative reasoning traces, thereby obviating the need for expensive post hoc filtering. First, we demonstrate that, using only the first 100 tokens out of reasoning trace of length up to 91k tokens, challenging examples can be reliably identified as those exhibiting higher loss at a randomly perturbed checkpoint of the pretrained model. We then establish that, during SFT—where model parameters undergo relatively small changes—examples with similar loss values over their first 1k tokens, measured at a small number of noisy checkpoints extrapolating along the fine-tuning direction, induce similar gradients throughout training. Our Token-Efficient Model Perturbation (TEMP)-based data selection method curates an SFT dataset that is both

[1]Department of Computer Science, University of California Los Angeles (UCLA), California, USA [2]Optum AI, UnitedHealth Group, Minneapolis, Minnesota, USA. Correspondence to: Hongyi Henry Jin <hongyi@cs.ucla.edu>.

*Proceedings of the 43rd International Conference on Machine Learning*, Seoul, South Korea. PMLR 306, 2026. Copyright 2026 by the author(s).

[1]Our repo is available at https://bigml-cs-ucla.github.io/TEMP-project-page/

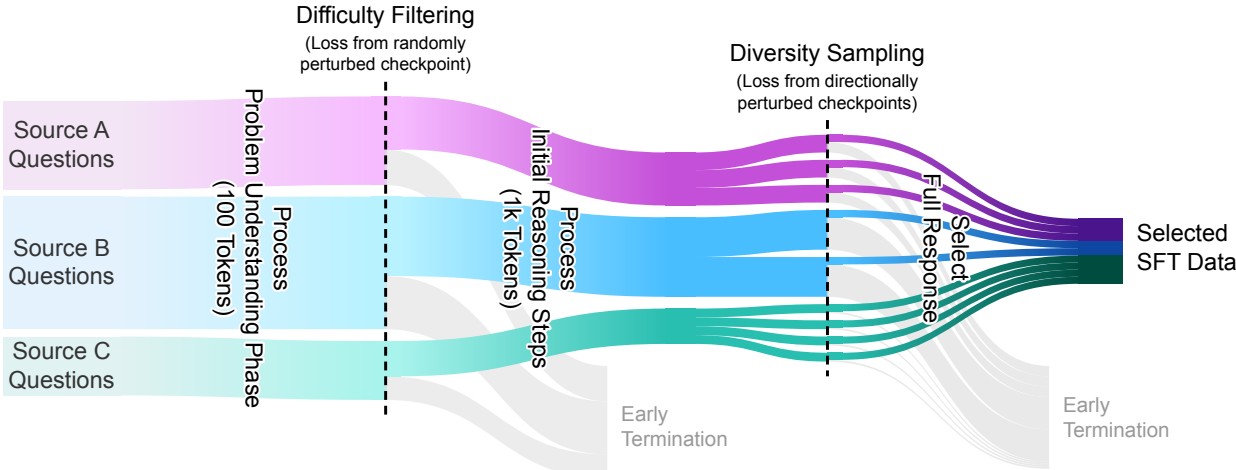

*Figure 1.* Overview of TEMP: Token-Efficient Model Perturbation Reasoning Data Selection. First, using only the first 100 tokens of reasoning traces, we identify challenging examples as those exhibiting higher loss at a randomly perturbed checkpoint of the pretrained model (Sec. 3.1). Then, we cluster examples based on their first 1k token loss values, measured at a small number of noisy checkpoints extrapolating along the fine-tuning direction, and sample a diverse subset (Sec. 3.2). The curated difficult and diverse SFT dataset enables efficient training of high-performing reasoning models.

diverse and challenging, enabling efficient training of high-performing reasoning models.

We conduct extensive experiments for fine-tuning Qwen2.5-7B (Qwen et al., 2025) and Llama3.1-8B (Dubey et al., 2024) on M23K medical reasoning (Huang et al., 2025) and OpenThoughts (Guha et al., 2025) datasets and show that our approach outperforms existing baselines by up to 1.7% while being 91% more token efficient.

## 2. Related works

LLM reasoning capabilities can be effectively induced using small, carefully curated sets of reasoning traces. This result has been verified in different settings, such as math reasoning (Muennighoff et al., 2025; Ye et al., 2025), medical reasoning (Huang et al., 2025), safety alignment (Wang et al., 2025), and instruction tuning (Li et al., 2024b; Liu et al., 2024) and with different model architectures (Zhao et al., 2025). In particular, difficulty and diversity are widely recognized as key attributes of high-quality reasoning data.

### 2.1. Data selection for Instruction Tuning

Fine-tuning on smaller but high-quality subsets of language data has been shown to substantially improve the performance of large language models (Li et al., 2024a), motivating a growing body of work on identifying and curating high-quality training data. Two properties are widely regarded as central to data quality: ***difficulty*** and ***diversity***. For instruction tuning, Zhou et al. (2023) manually curated a high-quality dataset spanning diverse topics and sources, demonstrating for the first time that fine-tuning on as few

as 1k examples can achieve state-of-the-art performance. Subsequent work has focused on more scalable approaches to data selection. For estimating difficulty, prior methods have employed LLM-based judges (Liu et al., 2024; Chen et al., 2024b), perplexity-based metrics (Li et al., 2024b), learnability measures based on loss reduction during training (Zhou et al., 2025), and gradient magnitude (Liu et al., 2025b). To promote diversity, both embedding-based methods (Liu et al., 2024; Bhatt et al., 2024b) and gradient-based methods (Zhang et al., 2025; Yang et al., 2024) have been explored, with gradient diversity shown to significantly outperform embedding diversity (Yang et al., 2024; Jung et al., 2025). However, these approaches become less effective for supervised fine-tuning reasoning models, where the extremely long chains of thought required for fine-tuning introduce substantial noise and variance into loss-, embedding-, and gradient-based signals. As a result, these metrics are dominated by trace length, verbosity, and stylistic variation rather than the underlying reasoning difficulty or informational content, limiting their usefulness for selecting high-quality reasoning data, as we confirm in our experiments.

### 2.2. Data selection for Reasoning Models

For reasoning models, Muennighoff et al. (2025) showed that for mathematical problems solving, supervised fine-tuning on 1k high-manually-curated difficult and diverse long reasoning traces can outperform state-of-the-art reasoning models, e.g. OpenAI's o1 (OpenAI, 2024) and DeepSeek R1 (Guo et al., 2025) trained with reinforcement learning on very large pools of data. Subsequent work prompts an LLM to estimate the difficulty of each question and retains only the hardest instances (Guha et al.,

```
Q1 (Easy question): Write the decomposition of the vector \(\vec{a}\) in terms of the vectors
    \[ \vec{p}=\{1,2,4\}, \quad \vec{q}=\{1,-1,1\}, \quad \vec{r}=\{2,2,4\}, \quad \vec{a}=\{-1,-4,-2\}. \]
A: Okay, so I need to decompose the vector \(\vec{a} = \{-1, -4, -2\}\) in terms of the vectors \(\vec{p} = \{1,
    2, 4\}\), \(\vec{q} = \{1, -1, 1\}\), and \(\vec{r} = \{2, 2, 4\}\).
-----------------------------
Q2 (Difficult Question): In a football tournament, each team is supposed to play one match against each of the other
    teams. However, during the tournament, half of the teams were disqualified and did not participate further.
    As a result, a total of 77 matches were played, and the disqualified teams managed to play all their matches
    against each other, with each disqualified team having played the same number of matches. How many teams
    were there at the beginning of the tournament?
A: Okay, let me try to figure out this football tournament problem. So, the question is: There was a tournament
    where each team was supposed to play against every other team once. But halfway through, half the teams got
    disqualified. In the end, a total of 77 matches were played. The important points are that the disqualified
    teams played all their matches against each other before being disqualified, and each disqualified team
    played the same number of matches. We need
```

*Figure 2.* Problem understanding phase in the first 100 response tokens. For the easy problem, the model only changes the format, which leads to low loss with low uncertainty. For the hard problem, it strategically identifies "the important points" which leads to higher loss.

2025), or consider response-length as a surrogate for difficulty (Huang et al., 2025; Guha et al., 2025) assuming that harder problems require more reasoning (Muennighoff et al., 2025; Li et al., 2025; Shen et al., 2025). For difficulty filtering, LLM-based filtering often outperforms response-length. For diversity, the data are categorized by an LLM oracle and a set of diverse samples is selected from the categories (Muennighoff et al., 2025; Huang et al., 2025). However, the extensive LLM-based filtering is very expensive and often yields suboptimal results.

We show that difficult and diverse reasoning traces can be identified based on initial CoT tokens, without prompting auxiliary LLMs, thereby substantially reducing the cost and improving accuracy of curating high-quality reasoning data.

## 3. Methods

In this section, we describe an efficient approach for selecting difficult and diverse reasoning traces from a data mixture $V$ containing $m$ sources, i.e., $V = \{V_1, \cdots, V_m\}$. Each data source $V_s$ contains different types of problems, e.g. different mathematical or medical problems.

In Sec. 3.1, we show that the very first part of the model response—*problem understanding phase*—can be used to quantify the difficulty of the problem. In Sec. 3.2 we show that the initial reasoning steps—*early solution phase*—can reliably quantify reasoning similarity. Figure 1 provides an overview of our Token-Efficient Model Perturbation (TEMP)-based method for reasoning data selection.

### 3.1. Filtering Difficult Problems

First, we show how problem difficulty can be efficiently estimated based on the initial part of the reasoning traces, without relying on external LLMs.

**Loss at a *randomly* perturbed checkpoint indicates difficulty.** Pretrained models often learn highly optimized, general-purpose feature extractors. As shown in Fig. 4,

geometrically, this places the model parameters in a wide basin corresponding to a low-loss solution. Evaluating data difficulty at the pretrained checkpoint is often ineffective because the model has already memorized or overfitted to complex outliers, resulting in deceptively low loss values across both easy and difficult exemplars. By introducing random perturbations to the model weights, this late-stage memorization is disrupted, thereby exposing the underlying geometry of the loss landscape. Easy, highly generalized examples reside within broad, flat minima and remain stable under noise, whereas inherently difficult, ambiguous, or anomalous examples occupy sharp, brittle minima. Consequently, weight perturbation acts as a stress test that induces sharp loss spikes exclusively for difficult data points, making the perturbed checkpoint an effective proxy for identifying structural data complexity and out-of-distribution instances.

**Loss of initial *response* tokens reflects problem difficulty.** However, the aggregate loss of either the problem statement or the full reasoning trace is a poor proxy for example difficulty. Many hard reasoning problems have concise, generic prompts whose intrinsic complexity remains latent before the model starts responding to the prompt. Conversely, calculating the loss over complete reasoning traces introduces substantial noise, as the metric becomes heavily dominated by token-level stochasticity and superficial stylistic variations. This compounding variance effectively obscures the distinct loss signals generated by the truly critical and structurally complex deductive steps.

Next, we show that the loss of the initial part of the model response—where model starts understanding the problem and planning how to solve it—is a reliable indicator for problem difficulty. As shown in Figure 2, LLMs start their response by strategically summarizing the key points in the problem and anchor their reasoning on that. This ***problem understanding phase*** translates the raw prompt into a structured, internal mental model necessary for multi-step inference.

At a perturbed checkpoint, the loss calculated specifically over this rephrasing segment serves as an excellent indi-

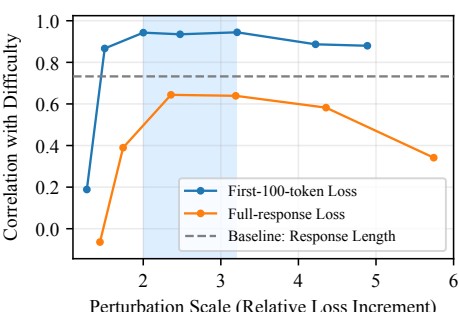

*Figure 3.* The correlation of different heuristics with difficulty. We label the difficulty of the problems in the m23k dataset with Gemini-2.5-flash-lite, and compare how well the difficulty correlates with metrics including response length and loss. The loss on the initial 100 tokens (corresponding to problem understanding phase) on the perturbed checkpoints have higher correlation with difficulty than longer responses, and bringing the loss around 2x∼3x higher than pretrained model loss gives the best correlation.

cator of difficulty because it measures the stability of the model's conceptual comprehension. For difficult problems, injecting weight noise causes a significant loss spike during the rephrasing process, exposing the model's underlying uncertainty about complex problems before it even attempts the downstream deduction.

Figure 3 shows the loss of the initial 100 tokens of the reasoning traces, evaluated at the randomly perturbed checkpoint $\theta_{rnd}$ correlates well with problem difficulties (evaluated by Gemini-2.5-flash-lite), better than the heuristic response length baseline. Formally, we define:

$$\mathcal{L}_i^{pr}(\theta_{rnd}) = \sum_{j=1}^{h} -\log p(x_j|x_{<j}, \theta_{rnd}), \qquad (1)$$

$$\theta_{rnd} = \theta_0 + \lambda \cdot \xi \quad \text{s.t.} \quad \frac{\mathcal{L}_i^{pr}(\theta_{rnd})}{\mathcal{L}_i^{pr}(\theta_0)} \in [2,3]$$

where $\theta_0$ is the pretrained checkpoint and $\xi \in \mathcal{N}(0, I)$ is a Gaussian noise. Then Figure 3 shows that $\mathcal{L}_i^{pr}(\theta_{rnd})$ is a good indication of the difficulty. Empirically, we find that $h \in [100, 120]$ works well, and we use $h = 100$ in our experiments. Using Qwen3.5-9B as a judge, the average length of this problem understanding phase has a mean of 95.4 tokens on the M23k dataset. We provide more ablation on the choice of $h$ in the appendix.

**Sampling a difficult problem mixture from data sources.** Next, we discuss how to filter out the easy problems in each data source $V_s \subseteq V$ and how many difficult examples to sample from each source $V_s$.

***Step 1. filtering out easy problems in each source.*** First, we filter out easy examples in each source. To do so, we cluster examples in each source $i \in V_s$ based on their initial token losses $\mathcal{L}_i^{pr}(\theta_{rnd})$. This partitions examples in each

source into two groups, namely easy $V_s^e$ and difficult $V_s^d$. Examples in $V_s^e$, which has lower average loss, is discarded.

***Step 2. sampling more from difficult and brittle sources.*** Next, we use the calculated initial token losses, $\mathcal{L}_i^{pr}(\theta_{rnd})$, to quantify *inherent and brittle difficulty* of remaining problems in every source $V_s^d \subseteq V_s$, and sample more from sources containing more difficult problems.

Inherently difficult problems have a high value of $\mathcal{L}_i^{pr}(\theta_{rnd})$ at the perturbed checkpoint, indicating that they require multi-step logical deduction, advanced mathematical abstraction, or highly counterintuitive causal relationships that the model's architecture has failed to internalize or generalize, independent of any optimization noise. Formally, we define **inherent difficulty** of a source $s$ as:

$$d_s^{in}(\theta_n) = \mathbb{E}_{i \in V_s^d}[\mathcal{L}_i^{pr}(\theta_{rnd})]. \qquad (2)$$

Brittle difficulty captures a class of data points where a model achieves a deceptively low loss at the pretrained checkpoint but suffers a catastrophic loss spike under weight perturbation, signaling memorization rather than conceptual understanding and robust generalization. This category primarily consists of borderline complexity tasks that sit right at the edge of the model's capabilities, requiring absolute, fragile precision across its weights to output correctly. We define **brittle difficulty** of a source $s$ as:

$$d_s^{br} = \mathbb{E}_{i \in V_s^d}[\mathcal{L}_i^{pr}(\theta_{rnd}) - \mathcal{L}_i^{pr}(\theta_0)]. \qquad (3)$$

For $m$ data sources and a total budget of $N$, we select $N_s$ reasoning traces from source $s$, based on the softmax of the geometric mean of the source inherent $d_s^{in}$ and brittle difficulty $d_s^{br}$:

$$N_s = N \cdot \frac{\exp(d_s)}{\sum_{j=1}^m \exp(d_s)}, \quad d_s = \sqrt{d_s^{in} d_s^{br}}. \qquad (4)$$

If a source contains less data than the budget, the remaining budget is distributed proportionally over remaining sources.

### 3.2. Sampling Diverse Reasoning Traces

Next, we discuss finding a subset of $N_s$ difficult examples from each source with diverse gradients during fine-tuning.

**Loss geometry is smooth along the fine-tuning direction.** During fine-tuning, the model primarily re-weights or combines existing features learned during pretraining, rather than learning entirely new ones. Therefore, fine-tuning dynamics are often well approximated by local (Gururangan et al., 2020; Tanwar et al., 2025), near-linear behavior around the pretrained initialization, with limited curvature (Fort et al., 2020). The low-dimensional subspace required for fine-tuning aligns smoothly with the flat, low-curvature directions already present at the pretrained checkpoint. Therefore, the mathematical properties governing the

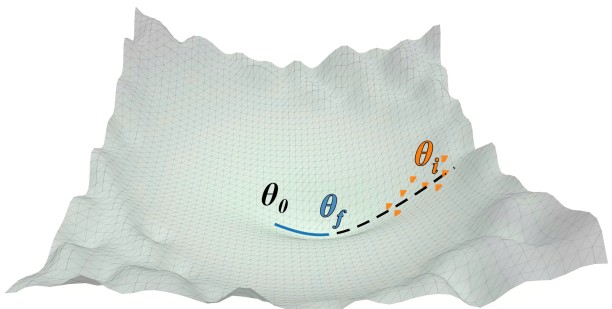

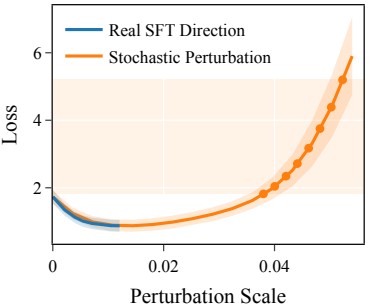

*Figure 4.* Schematic illustration of the wide basin containing the pretrained model parameters and fine-tuning trajectory. Fine-tuning proceeds in a low-dimensional flat direction of the basin (blue trajectory). Since the fine-tuning loss is very flat, examples remain poorly separated in loss space. On the other hand, extrapolated checkpoints in the fine-tuning direction (orange) lead to substantially better separation.

*Figure 5.* Illustration of the flat basin containing the pretrained Qwen2.5-7B-Instruct and its fine-tuning on M23k medical reasoning dataset. Extrapolated perturbations of the pretrained model along the fine-tuning direction effectively lie on the extended fine-tuning trajectory. Thus, they preserve the overall optimization geometry during fine-tuning but substantially better separate training examples based on their loss values.

loss in the fine-tuning direction remain bound to the original pretraining geometry (Aghajanyan et al., 2021).

Fig. 5 confirms that the loss landscape of Qwen2.5-7B is similar—flat parabola with low curvature—around the pretrained model and when fine-tuned for medical reasoning.

**Extrapolative perturbations in fine-tuning direction approximates loss level sets.** Since the fine-tuning loss landscape is flat and has low curvature, the relative properties of training examples—such as their individual loss values and gradients—change only slightly during fine-tuning. As a result, examples remain poorly separated in loss space, making it difficult to distinguish groups of similar examples.

To amplify this separation, we perturb the pretrained checkpoint along the fine-tuning direction, moving toward a nearby region where the loss remains smooth but exhibits higher curvature. Checkpoints in this perturbed region approximately trace level sets around the flat fine-tuning basin, while preserving the overall optimization geometry during fine-tuning. This is illustrated in Fig. 4. Importantly, losses in this region become more sensitive to differences between examples, leading to substantially better separation and discrimination among training samples.

*Directionally perturbed checkpoints.* For a given pretrained and fine-tuned model checkpoints $\theta_0, \theta_f$, we generate a set of $n$ perturbed checkpoints $\Theta$ extrapolating the fine-tuning direction $v$:

$$\Theta = \{\theta_j\}_{j=1}^n, \quad \theta_j = \theta_{j-1} + \lambda \cdot (1 + \xi_j) \odot v, \quad (5)$$

where $\xi_j \sim \mathcal{N}(0, I)$ is a standard Gaussian noise, $\lambda$ is the step size scaling factor, $\odot$ is the elementwise product, and $v = \theta_f - \theta_0$ is the vector representing the fine-tuning direction. We set $\lambda$ such that the perturbed checkpoints are evenly spaced in the distance from $\theta_0$ and the loss of the first and last checkpoints are substantially different. Adding

smaller directional perturbations to the previously perturbed checkpoint captures the loss landscape better than adding larger directional perturbations to the pretrained checkpoint.

Empirically, we find that an effective perturbation scheme yields, in expectation, approximately 1.05× higher loss for $\theta_1$ and 3× higher loss for $\theta_n$.

**Loss of initial *reasoning steps* reflects full reasoning loss.** Next, we calculate the loss of every example at the perturbed checkpoints. Fig. 6 shows that the loss of the first 1k tokens of reasoning traces are highly correlated with the loss of the full reasoning traces. Notably, this ***initial problem solving phase*** corresponds to when the model starts solving the problem step by step, and should not be confused with the *problem understanding phase* where the model starts understanding the problem and planning the solution in the first 100 response tokens. Therefore, we only need to calculate the loss of the first 1k tokens of each reasoning trace at perturbed checkpoints. We denote this loss as:

$$\mathcal{L}_i^{rs}(\theta) = \sum_{j=1}^r -\log p(x_j | x_{<j}, \theta), \quad r = 1k. \quad (6)$$

***Perturbed loss vector.*** For a set of perturbed checkpoints $\Theta = \{\theta_j\}_{j=1}^n$ defined in Eq. 5, perturbed loss trajectory of example $i$ is:

$$L_i = [\mathcal{L}_i^{rs}(\theta_1), \cdots, \mathcal{L}_i^{rs}(\theta_n)], \quad \theta_1, \cdots, \theta_n \in \Theta. \quad (7)$$

Next, we prove that examples with bounded difference in their perturbed loss vector have bounded gradient differences during fine-tuning.

**Perturbed loss vector differences bound gradient differences.** In the smooth and nearly flat loss landscape spanned by the perturbed checkpoints, pairwise differences between perturbed loss vectors upper-bound pairwise gradient differences during fine-tuning. In particular, in the wide basin

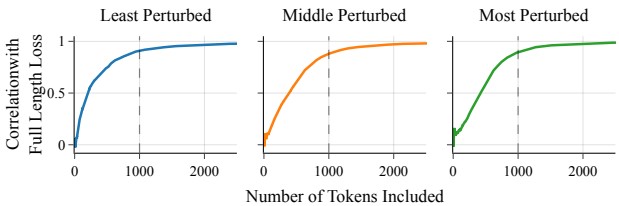

*Figure 6.* The Pearson Correlation between the mean loss of Qwen-2.5-7B-Instruct on initial and full reasoning tokens of the m23k dataset. Loss of the 1k initial reasoning tokens is highly correlated with the full response, across various perturbation strengths.

around the pretrained model, the gradient of the loss with respect to model parameters varies smoothly as a function of both the parameters and the data. As a result, two training examples that exhibit similar loss values at a small number of perturbed checkpoints will induce similar local loss geometries, and hence similar gradients throughout fine-tuning. This implies that the norm of the gradient difference between two examples can be upper bounded by a function of their loss differences evaluated at these perturbed checkpoints, with the bound governed by local smoothness constants of the loss. Next, we formalize this intuition.

**Theorem 3.1.** *Consider fine-tuning a pretrained model, where the curvature and gradient norms are upper-bounded by $C_H$, $G$, respectively; and parameter updates remain within an $\epsilon$-neighborhood of the pretrained initialization.*

$$|\mathcal{L}_{z_1}(\theta_j) - \mathcal{L}_{z_2}(\theta_j)| \leq \delta, \quad \forall j \in \{1, \ldots, n\},$$

*then the two examples have bounded gradient difference at any point $\theta$ reached during fine-tuning:*

$$|\langle \nabla \mathcal{L}_{z_1}(\theta) - \nabla \mathcal{L}_{z_2}(\theta), v \rangle| \leq \left(\frac{2\delta}{\lambda} + G\right)\left(\frac{1}{\sqrt{2}} + \tau\right) + C_H \epsilon,$$

*where $\lambda$ is the step size defined in Eq. 5.*

The proof is in Appendix A. We see that for a fixed $\delta$, when perturbed checkpoints are spaced further from each other, i.e., $\lambda$ is larger, tighter bounds on the gradient differences can be obtained. This clearly shows the benefit of perturbed checkpoints extrapolating the fine-tuning direction over those obtained from the fine-tuning trajectory.

**Sampling diverse reasoning traces from data sources.**
Theorem 3.1 shows that pairwise difference between perturbed loss vectors upper-bound pairwise gradient differences during fine-tuning. Based on this results, we cluster difficult examples in each source $V_s^d \subseteq V_s$ based on their perturbed loss vectors. Then, we sample examples with higher brittle difficulty for their initial 1k reasoning tokens.

***Step 1. clustering perturbed loss vectors in each source.***
Based on the above observation, we cluster the perturbed loss trajectories within each source. This yields groups of

---

**Algorithm 1** TEMP: Token-Efficient Model Perturbation for Reasoning Data Selection

---

**Require:** Data mixture $V = [V_1, \ldots, V_m]$, budget $N$, pretrained and fine-tuned model checkpoints $\theta_0, \theta_f$
**Ensure:** Curated SFT data $S$
 1: Perturb $\theta_0$ randomly to get $\theta_{rnd}$
 2: Perturb $\theta_0$ along $\theta_f - \theta_0$ to get $\Theta = \{\theta_i\}_{i=1}^n$ (Eq. 5).
 3: **for** each source $s \in [1, \ldots, m]$ **do**
 4:     Calculate $\mathcal{L}_i^{pr}(\theta_{rnd})$ for $i \in V_s$ (Eq. 1)
 5:     Cluster $\mathcal{L}_i^{pr}(\theta_{rnd})$ into two groups and keep difficult examples $V_s^d$ in cluster with higher mean $\mathcal{L}_i^{pr}(\theta_{rnd})$ loss
 6:     Calculate perturbed loss vectors $L_i$ for $i \in V_s^d$ using $\Theta$ (Eq. 7)
 7:     Cluster perturbed loss vectors and sample $N_s$ brittle examples to get $S_s$ (Eq. 4, 8)
 8: **end for**
 9: **return** $S = \bigcup_{s=1}^m S_s$

---

reasoning traces with bounded gradient differences during fine-tuning. For each source, we partition the data into $\lfloor N_s/k \rfloor$ clusters, where $N_s$ is the budget for each source according to Eq. 4, and $k$ is a hyperparameter to control how many data to take from each cluster. Empirically, the performance is not susceptible to the selection of $k$ as shown in Figure 12, and we use $k = 4$ in all our experiments.

***Step 2. sampling brittle reasoning traces from clusters.***
Then, we sample $k$ data points with the highest brittle reasoning difficulty from each cluster:

$$S_s^* = \arg\max_{\substack{S \subseteq V_s^d, \\ |S| \leq N_s}} \sum_{i \in S} \mathcal{L}_i^{rs}(\theta_n) - \mathcal{L}_i^{rs}(\theta_1). \tag{8}$$

Such examples consists of ambiguous or overconstrained prompts where the model has memorized an arbitrary resolution path, examples that contradict general logical priors, borderline complexity tasks that sit at the absolute limit of the model's capacity, and out-of-distribution stylistic formats where the low baseline loss merely reflects the superficial mimicry of rare template structures rather than true conceptual understanding.

The final sampled subset contains high-quality examples for supervised fine-tuning reasoning models.

The pseudocode of our method is illustrated in Alg. 1.

## 4. Empirical evaluation

In this section, we evaluate the effectiveness of our method for curating medical and mathematical reasoning datasets. We first discuss our experimental setting and then present our results, followed by an ablation study of different components of our TEMP method.

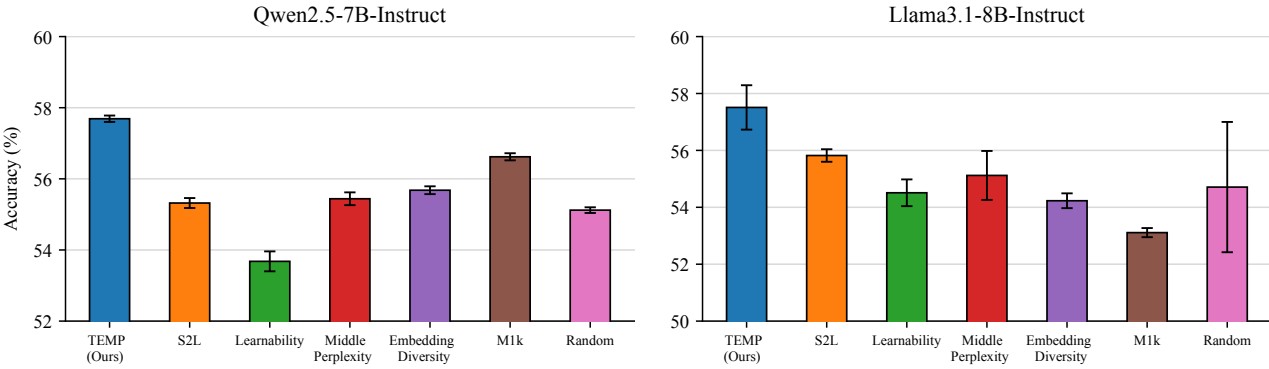

*Figure 7.* Our method, TEMP, outperforms all baselines when fine-tuning Qwen2.5-7B-Instruct (left) and Llama3.1-8B-Instruct (right) on 1k examples selected from the M23k medical reasoning dataset. Average accuracy is shown across 10 medical benchmarks.

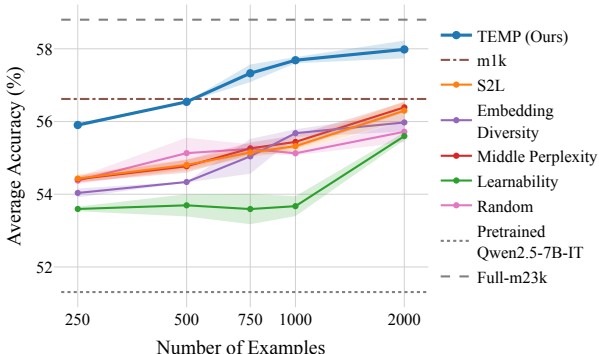

*Figure 8.* Fine-tuning Qwen2.5-7B-Instruct on on the M23k medical reasoning dataset. Average accuracy of subsets of various sizes is shown across 10 medical benchmarks. TEMP outperforms baselines across different budgets.

## 4.1. Experimental setup

**Datasets.** We apply our method to two datasets, M23k (Huang et al., 2025) and OpenThoughts-114k-math-correct (Hugging Face, 2025) (which we refer to as OpenThoughts-Math). Both datasets consist of question-answer pairs that require multi-step reasoning, along with reasoning traces distilled from DeepSeek-R1 (Guo et al., 2025). M23k contain 23,493 medical multiple-choice questions collected across 4 sources. The dataset has been through decontamination and deduplication, and easy questions that can be answered correctly by either Qwen2.5-7B-Instruct or Qwen2.5-32B-Instruct (Qwen et al., 2025) have been removed. Further, the distilled answers have been compared with ground truth answers, and reasoning with incorrect final answers have been removed to ensure the correctness of the final answer. OpenThoughts-Math is the math subset of OpenThoughts-114k (Guha et al., 2025) containing 56,370 questions from 4 sources. The dataset has been extensively filtered for difficulty and diversity using heuristic LLM-filtering methods. Similar to M23k, the model's generated results have been

verified, and incorrect answers have been removed.

**Experimental setting.** For SFT, we adopt the same hyperparameters in Huang et al. (2025) and use a small batch size of 16 to train for 5 epochs. For evaluation, we used the same benchmarks in Huang et al. (2025); Guha et al. (2025) respectively. Specifically, for m23k, we evaluate on 3 in-distribution and 7 out-of-distribution test sets, and report the average accuracy across all 10 test sets. For OpenThoughts-Math, we evaluate the model performance on 4 mathematical reasoning benchmarks. Additional details have been included in Section B.

## 4.2. Baselines

In this section, we show the effectiveness of our method compared to existing data selection methods, originally developed for instruction tuning as well as LLM-filtering methods. We consider the following baselines, which are representative of the difficult and diverse data selection methods described in Sec. 2:

**Learnability** (Zhou et al., 2025) selects examples with the largest reduction in loss during fine-tuning.

**Middle Perplexity** (Marion et al., 2023) ranks examples by their finetuned model perplexity and selects those centered at the median.

**Embedding Diversity** (Bhatt et al., 2024a) clusters finetuned-model embeddings of examples via $k$-medoids and selects the centroids.

**S2L** (Yang et al., 2024) cluster examples based on their loss values during training a proxy model and samples equally from the clusters.

**M1K** (Huang et al., 2025) uses LLM-based difficulty and diversity filtering.

**Random** selects a subset of examples uniformly at random.

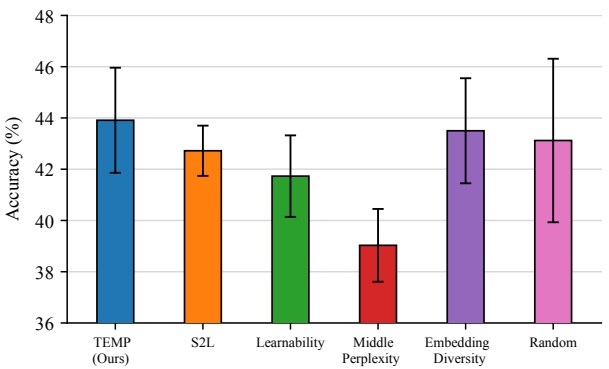

*Figure 9.* Comparison of our method, TEMP, with baselines for fine-tuning Qwen2.5-7B-Instruct on 1k examples selected from the highly-curated OpenThoughts-Math dataset.

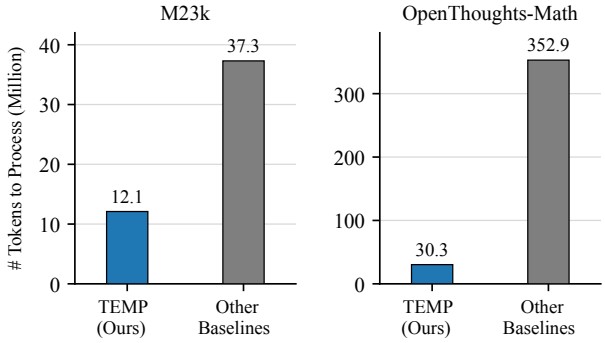

*Figure 10.* Token efficiency of our method (TEMP) vs baselines for selecting 1k data. While baselines requires processing the entire reasoning traces, TEMP only processes the initial tokens, improving token efficiency by 91% on the OpenThoughts-Math dataset.

### 4.3. Results

Next, we compare the performance of our method with baselines across different model architectures and data domains.

**Medical reasoning** Figure 7 shows that across both Qwen2.5-7B-Instruct and Llama-3.1-8B-Instruct, our method (TEMP) achieves the highest average accuracy for selecting 1k examples from M23k, outperforming the next-best baseline by up to 1.1% on Qwen2.5 and 1.7% on Llama3.1. Notably, other loss-based selection methods such as S2L achieve the second-best performance on Llama3.1, but fail to generalize to Qwen2.5, performing only marginally better than random selection. In contrast, our method ranks first across both model architectures, demonstrating its generalizability.

With Qwen2.5-7B-Instruct, we further vary the number of examples we select and present the results in Figure 8. Results show that our method scales well with the selection budget and consistently outperform the other baseline algorithms by a large margin.

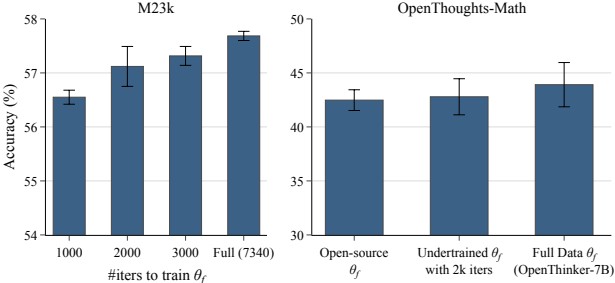

*Figure 11.* Performance of our method for selecting 1k data from **Left:** M23k dataset and **Right:** OpenThoughts-Math dataset, when using suboptimal $\theta_f$ for calculating $v$. Even when $\theta_f$ is is not exact, our method still obtains reasonable performance. However, using a significantly undertrained $\theta_f$ harms the performance.

*Table 1.* Sequential ablation of each component in our method. Each row removes one more component from the previous row.

| Diff. Filter. (Eq. 1) | Src. Budget Weight. (Eq. 4) | Diversity Samp. (Sec. 3.2) | Brittle Sel. (Eq. 8) | Acc. |
|---|---|---|---|---|
| ✓ | ✓ | ✓ | ✓ | 57.69% |
| ✓ | ✓ | ✓ | ✗ | 56.52% |
| ✓ | ✓ | ✗ | ✗ | 55.28% |
| ✓ | ✗ | ✗ | ✗ | 55.02% |
| ✗ | ✗ | ✗ | ✗ | 54.09% |

**Mathematical reasoning** Beyond medical reasoning, we evaluate our method on mathematical reasoning to further assess its generalizability across data domains. We fine-tune Qwen2.5-7B-Instruct on the 1k selected subset, and the results are shown in Figure 9. Consistent with the medical reasoning results, our method achieves the highest accuracy among all baselines, further confirming its generalizability across diverse data selection settings.

**Token efficiency** Figure 10 shows that despite its higher performance, our method is significantly more token-efficient than the baselines. This is because our method requires loss evaluation over substantially fewer tokens during data selection: approximately 3 times fewer in M23k and more than 10 times fewer in OpenThoughts-Math. This reduction stems directly from evaluating the loss only over initial reasoning tokens: easy examples are discarded after evaluating only 100 tokens, diversity clustering is performed after evaluating only 1k out of up to 91k reasoning tokens.

Together, these results demonstrate that our methods achieves superior performance while dramatically improving the token efficiency.

### 4.4. Ablation studies

**Sensitivity to the finetune direction $v$.** In Figure 11, we ablate the performance of our method when the finetune direction $v = \theta_f - \theta_0$ is inaccurate due to a suboptimal

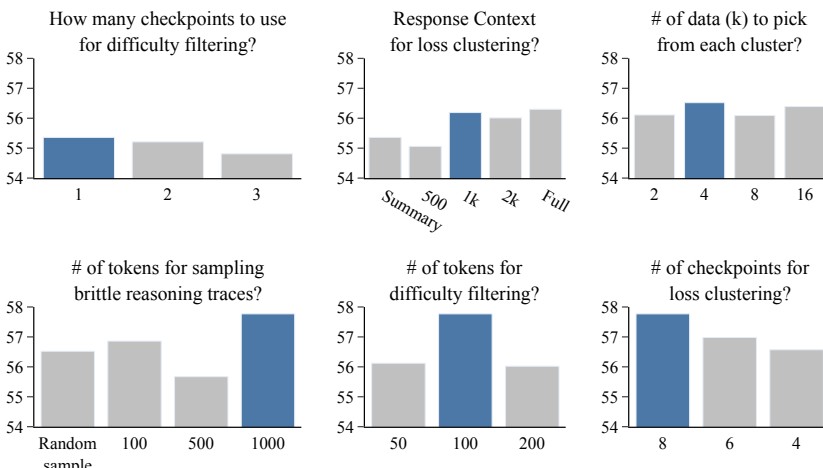

*Figure 12.* Ablation Study of the hyperparameters, highlighted bars indicates the hyperparameter used. **Top Left:** We tried using 1∼3 perturbed checkpoint for clustering in difficulty filtering. Using 1 or 2 perturbed checkpoints have a similar effect. **Top Right:** Number of tokens to sum over in Equation (6) for sampling brittle reasoning traces (Eq. 8). Using 1k tokens works the best here. **Bottom Left:** Definition of Eq. 6 when clustering. "Summary" is the solution the model outputs after reasoning. Using the first 1k tokens has similar performance as full response. **Bottom Right:** Number of data to select from each cluster. The selection of $k$ has a relatively small effect.

$\theta_f$. On the M23k dataset, apart from using the checkpoint finetuned on the entire available data with 7340 iterations, we use undertrained checkpoints that are only trained for 1000-3000 iterations. On the OpenThoughts-Math dataset, apart from using OpenThinker-7B as $\theta_f$, we consider an undertrained model trained with 2k iteration (corresponding to $1/10$ of the finetuning of OpenThinker-7B) and an open-source finetuned model for math reasoning, namely SynLogic-7B (Liu et al., 2025a), that is trained with a different recipe on a different dataset. Results show that even when $\theta_f$ is is not exact, our method still obtains reasonable performance. But, using a significantly undertrained model harms the performance. This confirms the importance of extrapolating the fine-tuning direction.

**Ablation on the effect of each component** We conduct ablation studies to understand the impact of different components of our method, and present the results in Table 1. We sequentially remove each component and observe a consistent drop in performance, confirming that each contributes meaningfully. As shown, removing any single component degrades performance, confirming that each contributes meaningfully. Removing brittle selection within clusters leads to a 1.17% drop in accuracy, and further removing diversity sampling leads to an additional 1.24% drop, demonstrating the importance of within-cluster example quality and gradient-diverse selection. Removing source budget weighting contributes a further 0.26% drop, showing the benefit of allocating more budget to harder sources. Finally, removing difficulty filtering leads to an additional 0.93% drop, confirming that filtering out easy examples early is essential for curating high-quality reasoning data.

**Sensitivity to hyperparameters** Figure 12 shows that our

method is largely robust to hyperparameter choices such as the number of perturbed checkpoints for difficulty filtering, the response context for loss clustering (where 1k tokens performs comparably to the full response), the number of data points selected per cluster ($k = 4$ works best), the number of tokens for difficulty filtering (100 tokens is optimal), and the number of checkpoints for loss clustering (8 checkpoints preferred). For sampling brittle reasoning traces, 1k tokens works best.

# 5. Conclusion

We demonstrated that high-quality supervised fine-tuning data for reasoning models can be curated efficiently by identifying difficult and diverse reasoning traces based on initial CoT tokens. By analyzing losses at a noisy perturbed checkpoint of the pretrained model, we showed that challenging examples can be reliably detected based on the initial 100 CoT tokens. We further established that examples exhibiting similar loss patterns over their first 1k reasoning tokens across a small number of perturbed checkpoints extrapolating along the fine-tuning trajectory provably induce similar gradients. Building on these insights, we introduced a lightweight pipeline that selects diverse and difficult reasoning traces using only the initial steps of reasoning. Extensive experiments on medical and mathematical reasoning benchmarks with Qwen2.5-7B and Llama3.1-8B confirm that our approach outperforms existing baselines by up to 1.7% while being 91% more token efficient.

## Acknowledgements

This research was supported by Optum AI Science, NSF CAREER Award 2146492, NSF-Simons AI Institute for Cosmic Origins (CosmicAI), and NSF AI Institute for Foundations of Machine Learning (IFML).

## Impact Statement

This paper presents work whose goal is to advance the field of Machine Learning. There are many potential societal consequences of our work, none which we feel must be specifically highlighted here.

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

## A. Proof for Theorem 3.1

We first formalize the assumptions needed for the theorem.

**Assumption A.1** (Local SFT). Let $\Theta \subset \mathbb{R}^d$ be the local region that SFT on a dataset $D$ can reach from initialization $\theta_0$. Let $L_z(\theta)$ be the loss of data $z \in D$ evaluated by model parameter $\theta \in \Theta$. We assume that during SFT $\Theta$ satisfies the following requirements:

1. During SFT, parameters remain in a local region of diameter at most $\epsilon$ around the pretrained initialization.

2. For any pair of examples $z_1, z_2 \in D$, the loss difference $L_{z_1}(\theta) - L_{z_2}(\theta)$ is locally well approximated by a quadratic function on $\Theta$, with curvature bounded by $C_H$ and gradient norm bounded by $G$.

3. The perturbation direction $e$ used in Eq. (5) is dense, i.e., $\|e\|_\infty^2 \le \mu/d$ for a moderate constant $\mu$.

We now prove Theorem 3.1.

**Theorem 3.1.** *Consider fine-tuning a pretrained model, where the curvature and gradient norms are upper-bounded by $C_H$, $G$, respectively; and parameter updates remain within an $\epsilon$-neighborhood of the pretrained initialization.*

$$|\mathcal{L}_{z_1}(\theta_j) - \mathcal{L}_{z_2}(\theta_j)| \le \delta, \quad \forall j \in \{1, \dots, n\},$$

*then the two examples have bounded gradient difference at any point $\theta$ reached during fine-tuning:*

$$|\langle \nabla \mathcal{L}_{z_1}(\theta) - \nabla \mathcal{L}_{z_2}(\theta), v \rangle| \le \left(\frac{2\delta}{\lambda} + G\right)\left(\frac{1}{\sqrt{2}} + \tau\right) + C_H \epsilon,$$

*where $\lambda$ is the step size defined in Eq. 5.*

*Proof of Theorem 3.1.* Let $f(\theta) = L_{z_1}(\theta) - L_{z_2}(\theta)$. We prove the claim for an adjacent pair of perturbed checkpoints whose realized direction is aligned with $v$. Without loss of generality, take $v = (\theta_f - \theta_0)/\|\theta_f - \theta_0\|$ to be the unit fine-tuning direction from Eq. (5). Write

$$m = \frac{\theta_j + \theta_{j-1}}{2}, \qquad u = \frac{\theta_j - \theta_{j-1}}{\|\theta_j - \theta_{j-1}\|}, \qquad s = \|\theta_j - \theta_{j-1}\|.$$

For the perturbation in Eq. (5), $s = \lambda\|(1 + \xi_j) \odot v\|$. Since $f$ is quadratic in the local region, the symmetric difference around $m$ cancels the quadratic term:

$$f(\theta_j) - f(\theta_{j-1}) = s\langle \nabla f(m), u \rangle.$$

By the loss-similarity assumption, $|f(\theta_j) - f(\theta_{j-1})| \le 2\delta$, so

$$|\langle \nabla f(m), u \rangle| \le \frac{2\delta}{s}.$$

Decompose the fine-tuning direction as $v = \langle v, u \rangle u + w$ with $w \perp u$. For slack $\tau$, the perturbation construction yields $\|w\| \le \sqrt{\frac{1}{2} + \sqrt{2}\tau - \tau^2}$. Using $\|\nabla f(m)\| \le G$,

$$|\langle \nabla f(m), v \rangle| \le \frac{2\delta}{s} + G\sqrt{\frac{1}{2} + \sqrt{2}\tau - \tau^2}.$$

It remains to transfer the bound from $m$ to any $\theta$ reached during fine-tuning. Since $\theta$ and $m$ lie in the same local region of diameter $\epsilon$ and $\|\nabla^2 f\|_{\text{op}} \le C_H$,

$$|\langle \nabla f(\theta) - \nabla f(m), v \rangle| \le C_H \epsilon.$$

Combining the two inequalities gives

$$|\langle \nabla f(\theta), v \rangle| \le \frac{2\delta}{\lambda\|(1 + \xi_j) \odot v\|} + C_H \epsilon + G\sqrt{\frac{1}{2} + \sqrt{2}\tau - \tau^2}.$$

Finally, for dense perturbation directions in high dimension, the noisy perturbation norm is concentrated around $\sqrt{2}$. Let $Z = \|(1 + \xi_j) \odot v\|^2$; then $\mathbb{E}[Z] = 2$. If $\|v\|_\infty^2 \le \mu/d$, then Chebyshev's inequality gives

$$\Pr\Big(\|(1 + \xi_j) \odot v\|^2 < (\sqrt{2} - \tau)^2\Big) \le \frac{6\mu}{d(2\sqrt{2}\,\tau - \tau^2)^2}.$$

Thus, with probability at least $1 - \frac{6\mu}{d(2\sqrt{2}\,\tau - \tau^2)^2}$,

$$|\langle \nabla L_{z_1}(\theta) - \nabla L_{z_2}(\theta), v \rangle| \le \frac{2\delta}{\lambda(\sqrt{2} - \tau)} + C_H \epsilon + G\sqrt{\frac{1}{2} + \sqrt{2}\,\tau - \tau^2}.$$

For $\tau \le 1/\sqrt{2}$, we have $\frac{1}{\sqrt{2}-\tau} \le \frac{1}{\sqrt{2}} + \tau$ and $\sqrt{\frac{1}{2} + \sqrt{2}\,\tau - \tau^2} \le \frac{1}{\sqrt{2}} + \tau$, which yields the bound in the theorem. $\qquad \square$

# B. Additional Experiment Setup

## B.1. SFT specification

All our training are carried out through the `llama-factory` (Zheng et al., 2024) framework. We used the same hyperparameters as the one used in Huang et al. (2025); Muennighoff et al. (2025). Specifically, we finetuned the Qwen2.5-7B-Instruct model (Qwen et al., 2025) for 5 epochs. We used learning rate $1 \times 10^{-5}$ with linear warmup for the first $5\%$ of all iterations, and then use cosine annealing to decrease the learning rate to 0. We used the AdamW_torch optimizer with $\beta_1 = 0.9, \beta_2 = 0.95$. Training sequence length is capped to 8k tokens on M23k dataset and 32k tokens (which is the maximum context length of Qwen2.5-7B) on OpenThoughts-Math.

## B.2. Baselines

**Learnability** (Zhou et al., 2025) The learnability score of a data sample is defined as the difference between the loss of the example w.r.t. the model before and after finetuning, namely $\theta_0$ and $\theta_f$.

**Middle Perplexity** (Marion et al., 2023) Middle perplexity sorts the data by its perplexity evaluated on the finetuned-model $\theta_f$, and then selects the consecutive data centered at the median. This removes the most perplexing data, which may contain error or is too difficult for the model to learn, and excludes the easiest data as well.

**Embedding Diversity** (Bhatt et al., 2024a) This method uses the last hidden state of the finetuned model $\theta_f$ averaged across all model response as the embedding for each answer, and cluster the embeddings with $k$-medoids algorithm. The medoids are selected.

**S2L** (Yang et al., 2024) S2L evaluated the loss on a set of model checkpoints saved during the actual finetuning, cluster the loss, and selects data equally from each cluster.

**M1K** (Huang et al., 2025) uses LLM-based difficulty and diversity filtering.

**Random** selects a subset of examples uniformly at random.

## B.3. Evaluation Benchmarks

**M23k**  We include 3 in-distribution benchmarks, namely MedMCQA (Pal et al., 2022), MedQA-USMLE (Jin et al., 2021), and PubMedQA (Jin et al., 2019), and 7 out-of-distribution benchmarks, namely MMLU-Pro (Medical) (Wang et al., 2024), GPQA (Medical) (Rein et al., 2024), Lancet and NEJM clinical case reports, MedBullets (Chen et al., 2024a), MedXpertQA (Zuo et al., 2025). Generation temperature is set to be 0.7, and the maximal new tokens to generate is capped at 8k.

**OpenThoughts-Math**  We include 4 benchmarks AMC23, AIME24, AIME25 (MAA, 2024), and MATH500 (Lightman et al., 2023). Generation temperature is set to be 0.7, and the maximal new tokens to generate is capped at 28k tokens to ensure sufficient tokens for the prompt.

## B.4. Method and Baseline

Unless specified otherwise, we use OpenThinker-7B as the $\theta_f$ on the OpenThoughts-Math dataset and the model finetuned on the full M23k data as $\theta_f$ on the OpenThoughts-Math dataset. Other baselines use the same $\theta_f$ except for S2L, where a

smaller model Qwen2.5-0.5B-Instruct is used for finetuning on both OpenThoughts-Math and M23k dataset.

## C. Detailed Experiment Results

In Table 2 thorugh Table 8 we provide the numerical values of the performance of TEMP compared with the baselines presented in Figure 7 through Figure 9.

*Table 2.* Performance of finetuning Qwen2.5-7B-Instruct on the m23k dataset when selecting 250 examples.

| Method | Avg | In distribution | | | Out Of Distribution | | | | | | |
|---|---|---|---|---|---|---|---|---|---|---|---|
| | | MedMCQA | MedQA | PubMedQA | MMLU-Pro | GPQA | Lancet | MedBul4 | MedBul5 | MedXpert | NEJM |
| Random | 54.4% | 58.1% | 67.3% | **77.0%** | 59.3% | 46.4% | 61.3% | 54.9% | 44.3% | 14.7% | 60.5% |
| | ±0.1% | ±0.4% | ±0.4% | ±0.4% | ±0.5% | ±1.5% | ±0.4% | ±1.6% | ±0.2% | ±0.0% | ±0.2% |
| Learnability | 53.6% | 56.1% | 65.6% | 73.1% | 58.7% | 47.5% | 61.7% | 52.6% | 47.2% | 13.4% | 60.0% |
| | ±0.1% | ±0.1% | ±0.8% | ±0.3% | ±0.2% | ±1.4% | ±1.3% | ±0.5% | ±1.5% | ±0.5% | ±1.7% |
| Middle Perplexity | 54.4% | 57.6% | 67.5% | 74.3% | 62.3% | 45.8% | 59.1% | 54.4% | 46.4% | **16.1%** | 60.6% |
| | ±0.1% | ±0.4% | ±0.3% | ±0.9% | ±0.8% | ±1.9% | ±1.1% | ±0.2% | ±2.6% | ±0.4% | ±0.7% |
| Embedding Diversity | 54.0% | 57.6% | 67.6% | 74.6% | 59.1% | 46.8% | 61.0% | 52.3% | 46.6% | 14.4% | 60.4% |
| | ±0.1% | ±0.1% | ±0.2% | ±0.1% | ±0.1% | ±0.4% | ±0.1% | ±1.0% | ±1.8% | ±0.0% | ±0.7% |
| S2L | 54.4% | 57.3% | **68.7%** | 73.7% | 58.4% | 46.4% | **62.1%** | 55.0% | 46.8% | 14.8% | 61.1% |
| | ±0.1% | ±0.1% | ±0.4% | ±0.2% | ±0.0% | ±1.8% | ±0.2% | ±0.8% | ±0.3% | ±0.5% | ±1.4% |
| Ours | **55.9%** | **58.1%** | 68.3% | 75.2% | **63.2%** | **48.2%** | 61.4% | **57.5%** | **47.9%** | 15.9% | **63.3%** |
| | ±0.1% | ±0.1% | ±0.2% | ±0.2% | ±0.3% | ±1.5% | ±0.5% | ±1.6% | ±1.8% | ±1.6% | ±0.2% |

*Table 3.* Performance of finetuning Qwen2.5-7B-Instruct on the m23k dataset when selecting 500 examples.

| Method | Avg | In distribution | | | Out Of Distribution | | | | | | |
|---|---|---|---|---|---|---|---|---|---|---|---|
| | | MedMCQA | MedQA | PubMedQA | MMLU-Pro | GPQA | Lancet | MedBul4 | MedBul5 | MedXpert | NEJM |
| Random | 55.1% | 57.3% | 68.6% | 75.1% | 62.6% | 45.4% | **62.4%** | 56.0% | 48.7% | 15.7% | 59.5% |
| | ±0.4% | ±0.5% | ±1.3% | ±0.1% | ±0.0% | ±1.8% | ±1.7% | ±1.1% | ±1.0% | ±1.0% | ±1.8% |
| Learnability | 53.7% | 56.5% | 64.9% | 73.7% | 60.2% | **46.8%** | 60.6% | 52.8% | 48.1% | 14.1% | 59.3% |
| | ±0.3% | ±0.5% | ±0.7% | ±0.9% | ±0.3% | ±2.2% | ±1.5% | ±1.4% | ±1.3% | ±0.4% | ±1.6% |
| Middle Perplexity | 54.8% | 57.9% | 68.5% | 74.5% | 62.8% | 44.6% | 60.2% | 54.1% | 49.4% | 15.4% | 60.4% |
| | ±0.2% | ±0.2% | ±1.5% | ±1.2% | ±0.1% | ±0.3% | ±0.0% | ±0.5% | ±1.9% | ±0.7% | ±0.3% |
| Embedding Diversity | 54.3% | 57.4% | 66.7% | 74.8% | 61.7% | 45.3% | 61.9% | 52.1% | 47.9% | 15.3% | 60.3% |
| | ±0.0% | ±0.0% | ±0.3% | ±0.1% | ±1.5% | ±0.4% | ±0.5% | ±0.5% | ±0.8% | ±0.3% | ±1.6% |
| S2L | 54.8% | 57.4% | 68.6% | 75.0% | 61.8% | 46.0% | 61.9% | 53.9% | 47.7% | 14.7% | 61.1% |
| | ±0.2% | ±0.4% | ±0.4% | ±0.0% | ±1.3% | ±2.7% | ±2.2% | ±0.6% | ±1.6% | ±0.7% | ±0.1% |
| Ours | **56.5%** | **58.7%** | **72.3%** | **76.0%** | **63.8%** | 46.2% | 61.4% | **59.4%** | **50.2%** | **16.1%** | **61.4%** |
| | ±0.0% | ±0.1% | ±0.3% | ±0.1% | ±0.4% | ±0.0% | ±0.7% | ±1.6% | ±0.8% | ±0.7% | ±1.0% |

## D. Pseudo code for edge case selection

In extreme cases, we might encounter situations where a source doesn't have enough data to select from given the budget in Equation (4), or when a cluster contains less data than the number of data to select from each cluster. For instance, in the M23k dataset, the source PubMedQA only contains 29 examples. In this case, we redistribute the extra budget to the remaining sources / clusters according to their budget proportion. Detailed pseudocode is given in Algorithm 2. Given this redistribution, we provide the complete accurate pseudocode for diversity sampling in Algorithm 3.

## E. Model details and prompts for LLM-as-a-judge

### E.1. Labelling the difficulty

We specify the details of difficulty labelling used in Figure 3. We labelled the difficulty of each question in M23k with `gemini-2.5-flash-lite`. We provide a promt containing metrics and examples to ensure consistent labelling as done in Guha et al. (2025). The rubric is designed so that the five levels contain approximately the same number of questions in the dataset, avoiding a difficulty scale dominated by a single broad category. We enforced JSON output and asked the model

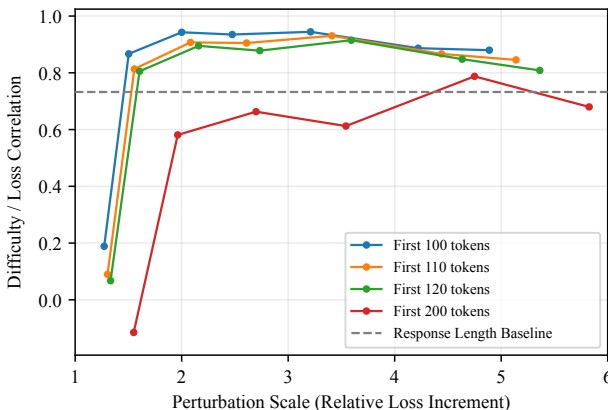

*Figure 13.* Additional results on the correlation of different heuristics with difficulty. Compared to including more prefix tokens, 100∼120 tokens have consistently high correlation with difficulty.

---

**Algorithm 2** RedistributeBudget: Constrained Allocation

---

**Require:** Total Budget $N$, Available Counts $\mathbf{n} = [n_1, \ldots, n_m]$, Weights $\mathbf{w} = [w_1, \ldots, w_m]$
**Ensure:** Allocations $\mathbf{c} = [c_1, \ldots, c_m]$
1: Indices $I \leftarrow [1, \ldots, m]$
2: Sort $I$ such that the ratio $n_i/w_i$ is non-decreasing
3: $N_{\text{rem}} \leftarrow N, \quad W_{\text{rem}} \leftarrow \sum \mathbf{w}$
4: **for** each index $i$ in sorted $I$ **do**
5:     $target \leftarrow N_{\text{rem}} \cdot (w_i/W_{\text{rem}})$
6:     **if** $n_i \leq target$ **then**
7:         $c_i \leftarrow n_i$
8:     **else**
9:         $c_i \leftarrow \lfloor target \rfloor$
10:     **end if**
11:     $N_{\text{rem}} \leftarrow N_{\text{rem}} - c_i, \quad W_{\text{rem}} \leftarrow W_{\text{rem}} - w_i$
12: **end forreturn c**

---

---

**Algorithm 3** Diversity Sampling

---

**Require:** Perturbed reasoning loss matrices with rows $L_i = [\mathcal{L}_i^{rs}(\theta_1), \ldots, \mathcal{L}_i^{rs}(\theta_n)]$ for $i \in V_s^d$ (Equation (7)), problem-difficulty values $\mathcal{L}_i^{pr}(\theta_{rnd})$ and $\mathcal{L}_i^{pr}(\theta_0)$ for $i \in V_s^d$, $s = 1, \ldots, m$, Clustering Algo $\mathcal{C}$, Targeted Size per Cluster $k$, Total Budget $N$

**Ensure:** Selected indices $S_{\text{final}}$

1: $S_{\text{final}} \leftarrow \emptyset$

  —— **Source Balancing** ——

2: $\mathbf{n} \leftarrow [|V_1^d|, \ldots, |V_m^d|]$          $\triangleright$ Available difficult examples per source

3: **for** each source $s \in [1, \ldots, m]$ **do**

4:      $d_s^{\text{in}} \leftarrow \text{Mean}_{i \in V_s^d}[\mathcal{L}_i^{pr}(\theta_{rnd})]$

5:      $d_s^{\text{br}} \leftarrow \text{Mean}_{i \in V_s^d}[\mathcal{L}_i^{pr}(\theta_{rnd}) - \mathcal{L}_i^{pr}(\theta_0)]$

6:      $w_s \leftarrow \exp(\sqrt{d_s^{\text{in}} d_s^{\text{br}}})$          $\triangleright$ Weights from Equation (4)

7: **end for**

8: $\mathbf{N}^{\text{src}} \leftarrow \textsc{RedistributeBudget}(N, \mathbf{n}, \mathbf{w})$

  —— **Loss Clustering** ——

9: **for** each source $s \in [1, \ldots, m]$ **do**

10:      $N_s \leftarrow \mathbf{N}_s^{\text{src}}$

11:      $K \leftarrow \max(1, \lfloor N_s/k \rfloor)$

12:      Clusters $\mathcal{K} = [C_1, \ldots, C_K] \leftarrow \mathcal{C}.\text{fit}(\{L_i\}_{i \in V_s^d}, K)$

13:      $\mathbf{n}' \leftarrow [|C_1|, \ldots, |C_K|]$          $\triangleright$ Vector of cluster sizes

14:      $\mathbf{w}' \leftarrow [1, \ldots, 1]$          $\triangleright$ Uniform weight vector (size $K$)

15:      $\mathbf{N}^{\text{clus}} \leftarrow \textsc{RedistributeBudget}(N_s, \mathbf{n}', \mathbf{w}')$

16:      **for** each cluster $j$ with allocation $c \in \mathbf{N}^{\text{clus}}$ **do**

17:          Sort $C_j$ by $\mathcal{L}_i^{rs}(\theta_n) - \mathcal{L}_i^{rs}(\theta_1)$ descending (Equation (8))

18:          $S_{\text{final}} \leftarrow S_{\text{final}} \cup \text{TopK}(C_j, c)$

19:      **end for**

20: **end for return** $S_{\text{final}}$

---

*Table 4.* Performance of finetuning Qwen2.5-7B-Instruct on the m23k dataset when selecting 750 examples.

| Method | Avg | In distribution | | | Out Of Distribution | | | | | | |
|---|---|---|---|---|---|---|---|---|---|---|---|
| | | MedMCQA | MedQA | PubMedQA | MMLU-Pro | GPQA | Lancet | MedBul4 | MedBul5 | MedXpert | NEJM |
| Random | 55.2% | 57.4% | 67.2% | **76.3%** | 62.4% | 46.0% | 62.1% | 54.5% | 49.5% | 15.7% | 61.2% |
| | ±0.1% | ±0.3% | ±0.5% | ±0.6% | ±0.1% | ±0.9% | ±1.0% | ±0.6% | ±1.8% | ±0.8% | ±1.2% |
| Learnability | 53.6% | 56.7% | 65.7% | 75.3% | 59.8% | 45.0% | 59.8% | 52.5% | 46.9% | 14.7% | 59.7% |
| | ±0.4% | ±1.4% | ±1.0% | ±0.5% | ±1.7% | ±0.5% | ±1.1% | ±1.8% | ±2.3% | ±0.3% | ±1.9% |
| Middle Perplexity | 55.3% | 56.7% | 69.6% | 74.9% | 62.1% | 45.9% | 59.1% | 57.5% | 48.7% | **16.4%** | **61.8%** |
| | ±0.2% | ±0.2% | ±0.6% | ±0.0% | ±1.4% | ±0.5% | ±0.1% | ±0.3% | ±0.3% | ±0.1% | ±0.6% |
| Embedding Diversity | 55.0% | 57.6% | 67.0% | 75.7% | **62.8%** | 47.1% | 62.3% | 56.7% | 48.2% | 15.0% | 58.1% |
| | ±0.5% | ±0.8% | ±0.4% | ±0.7% | ±0.2% | ±0.9% | ±1.8% | ±0.2% | ±3.4% | ±0.2% | ±0.9% |
| S2L | 55.1% | 57.3% | 69.2% | 76.0% | 61.8% | 48.2% | 63.0% | 54.2% | 48.1% | 14.6% | 59.1% |
| | ±0.1% | ±0.2% | ±0.3% | ±0.2% | ±0.8% | ±1.0% | ±1.6% | ±0.3% | ±2.6% | ±0.1% | ±1.4% |
| Ours | **57.3%** | **58.1%** | **70.9%** | 75.7% | 62.7% | **49.5%** | **64.9%** | **60.4%** | **53.7%** | 16.3% | 61.1% |
| | ±0.2% | ±0.5% | ±0.8% | ±0.1% | ±0.6% | ±0.3% | ±1.3% | ±0.6% | ±0.5% | ±0.2% | ±0.2% |

*Table 5.* Performance of finetuning Qwen2.5-7B-Instruct on the m23k dataset when selecting 1000 examples.

| Method | Avg | In distribution | | | Out Of Distribution | | | | | | |
|---|---|---|---|---|---|---|---|---|---|---|---|
| | | MedMCQA | MedQA | PubMedQA | MMLU-Pro | GPQA | Lancet | MedBul4 | MedBul5 | MedXpert | NEJM |
| Random | 55.1% | 57.7% | 69.0% | 75.0% | 62.7% | 46.9% | 61.2% | 54.9% | 49.8% | 15.1% | 58.9% |
| | ±0.1% | ±0.3% | ±0.5% | ±0.0% | ±0.1% | ±1.5% | ±1.7% | ±0.0% | ±1.8% | ±1.1% | ±1.5% |
| Learnability | 53.7% | 57.1% | 65.9% | 75.8% | 60.8% | 47.3% | 59.8% | 54.5% | 42.9% | 14.7% | 57.8% |
| | ±0.3% | ±0.3% | ±0.4% | ±0.9% | ±0.3% | ±1.4% | ±3.0% | ±1.3% | ±1.6% | ±0.2% | ±1.1% |
| Middle Perplexity | 55.4% | **58.5%** | 68.6% | 75.8% | 63.6% | 47.9% | 61.4% | 53.4% | 50.6% | 16.6% | 57.8% |
| | ±0.2% | ±0.2% | ±1.2% | ±1.6% | ±0.7% | ±1.8% | ±0.7% | ±0.8% | ±1.6% | ±0.0% | ±0.1% |
| Embedding Diversity | 55.7% | 57.5% | 67.3% | **76.6%** | 62.5% | 49.9% | **63.3%** | 58.3% | 47.2% | 15.8% | 58.5% |
| | ±0.1% | ±0.1% | ±0.9% | ±0.1% | ±1.2% | ±1.9% | ±0.2% | ±0.8% | ±3.1% | ±0.2% | ±1.1% |
| S2L | 55.3% | 58.0% | 68.5% | 76.4% | 62.5% | 47.9% | 62.6% | 55.2% | 47.4% | 15.1% | 59.5% |
| | ±0.1% | ±0.3% | ±0.2% | ±0.1% | ±0.5% | ±0.3% | ±0.7% | ±1.6% | ±1.3% | ±0.4% | ±0.7% |
| Ours | **57.7%** | 58.1% | **72.3%** | 76.5% | 63.0% | 49.6% | 61.4% | **61.5%** | 55.5% | **18.1%** | 60.8% |
| | ±0.1% | ±0.4% | ±0.7% | ±0.3% | ±0.5% | ±0.6% | ±0.2% | ±0.5% | ±0.6% | ±0.3% | ±0.2% |
| m1k | 56.6% | 57.3% | 70.4% | 74.9% | **64.5%** | **51.8%** | 60.4% | 58.8% | 51.4% | 15.3% | **60.8%** |
| | ±0.1% | ±0.2% | ±0.2% | ±0.2% | ±0.9% | ±1.5% | ±1.0% | ±1.2% | ±0.9% | ±0.8% | ±0.1% |

to slightly explain before generating the score.

After labelling the problems, we evaluate the loss on perturbed checkpoints and group the loss by the difficulty level. We calculate the Pierson correlation between the average loss of each difficulty level and the difficulty level (from 1 to 5) to plot Figure 3.

### E.2. Deciding boundary for Problem Understanding Phase

We use few-shot examples to prompt Qwen3.5-9B model to decide what is the boundary of the initial problem understanding phase. We show the model the problem statement and the first 500 tokens of the response, and ask the model to output verbatim the initial problem rephrasing. Since Qwen3.5 series share the same tokenizer as Qwen2.5 series, we count the number of tokens it extracted. Results show that M23k dataset's problem understanding phase has a mean of 95.4 tokens and an median of 54 tokens. The judge prompt and conversation layout are given in Figures 16 to 18.

The judge is invoked as a multi-turn chat: one system message (`JUDGE_SYSTEM_PROMPT`), four fixed few-shot demonstrations, and a final user turn for each target example. Each few-shot user turn supplies a medical multiple-choice question and the first 500 tokens of the model's `think` block; the corresponding assistant turn returns only a `<rephrase>` span copied verbatim from that prefix. On the target turn the model applies the same rules to the held-out question and prefix.

```
You will be given a medical question or clinical vignette. Your job is to grade the difficulty level according to
    the standard of medical education and clinical practice.

CRITICAL GUIDELINE: Do NOT grade based on Subject. Grade based on SPECIFICITY and COMPLEXITY.

Use this 5-Point Scale:

Level 1: Foundations, Anatomy & Physiology.
Basic definitions, anatomical locations, normal physiological functions, and general concepts of health vs. disease.
Examples: "Physiological role of Red Blood Cells," "Anatomical location of the Liver," "Defining Type 2 Diabetes,"
    "Normal Heart Rate range."

Level 2: Clinical Recognition & Standard Protocols.
Diagnosing conditions from classic presentations (no ambiguity) AND applying standard clinical rules (screening,
    prophylaxis, basic triage).
Examples: "Diagnosing Chickenpox from classic rash," "Identifying symptoms of the Common Cold," "Routine blood
    pressure screening."

Level 3: Standard Management & Basic Labs.
Identifying first-line treatments for common conditions OR interpreting standard lab values (CBC, Electrolytes).
Examples: "Prescribing inhalers for Asthma," "Treating dehydration with fluids," "Identifying high Glucose levels."

Level 4: Systemic Pathophysiology (Macroscopic).
Explaining the broad functional changes or causal chains underlying a condition at the organ/system level.
Examples: "Mechanism of Autoimmune destruction in Type 1 Diabetes," "Hemodynamic changes in Heart Failure."

Level 5: Advanced Mechanisms, Toxicology & Nuance.
Explaining specific molecular targets/receptors, Specific Toxicology, Complex Diagnoses, OR Management with
    Constraints.
Examples: "Mechanism of Digoxin (Na/K ATPase)," "Serotonin Syndrome," "Diagnosing Lupus," "Managing Asthma in
    Pregnancy."
```

*Figure 14.* System prompt for labelling M23k difficulty.

```
Question:
{question}

Please output your response in the following JSON format:
{
  "reasoning": "Brief explanation of the difficulty assessment.",
  "score": <integer_score>
}

Enforce json output.
```

*Figure 15.* User prompt for labelling M23k difficulty.

```
[system] JUDGE_SYSTEM_PROMPT
[user] few-shot example 1 (question + think prefix)
[assistant] <rephrase>...</rephrase>
[user] few-shot example 2 (question + think prefix)
[assistant] <rephrase>...</rephrase>
[user] few-shot example 3 (question + think prefix)
[assistant] <rephrase>...</rephrase>
[user] few-shot example 4 (question + think prefix)
[assistant] <rephrase>...</rephrase>
[user] target example (question + think prefix) <- model generates here
```

*Figure 16.* Conversation structure for the rephrase-boundary judge (Qwen3.5-9B).

```
You identify the **repetition / rephrasing prefix** at the start of a model's
'think' block on medical multiple-choice questions.
## Definition
**Repetition / rephrasing** = the initial span that restates the question, walks through
the clinical vignette in plain language, lists answer options, or sets up the task. It
does NOT yet apply medical reasoning.
**Still rephrase (include):**
- Meta openers: "Okay, let me work through this case"
- Paraphrasing the vignette: history, vitals, exam, labs, options
- Light parenthetical labels while listing facts: "(tachycardia)", "(a bit low)"
- Organizing facts: "Let me check her vitals: ..."
**NOT rephrase (stop before these):**
- Diagnostic inference: "points to PE", "suggests eclampsia", "raises concern for"
- Hypothesis generation: "Could she actually be pregnant?", "So, first thoughts"
- Analysis pivots: "Wait,", "Hmm,", "But wait,", "However, her symptoms"
- Knowledge retrieval for reasoning: "Let me recall what...", "First, let me think about each option"
- Differential / option elimination
## Critical rules
1. Copy the rephrase span **verbatim** from the think prefix -- character-for-character.
2. Stop at the **first** reasoning marker, even if the think prefix continues with
many paragraphs of reasoning afterward.
3. The think prefix is only the first 500 tokens and may contain lots of reasoning
**after** the rephrase. Do NOT extend rephrase to fill the prefix.
4. If there is no rephrase (reasoning starts immediately), output an empty rephrase.
## Output format (exactly)
<rephrase>
[verbatim rephrase text, or empty]
</rephrase>
Output nothing else.
```

*Figure 17.* System prompt (JUDGE_SYSTEM_PROMPT) for identifying the problem-understanding boundary.

```
Question:
{question}

Think prefix (first 500 tokens):
{think_prefix}
```

*Figure 18.* User message template for each few-shot and target example. Placeholders are filled with the MCQ stem and the truncated think block from the reasoning trace.

*Table 6.* Performance of finetuning Qwen2.5-7B-Instruct on the m23k dataset when selecting 2000 examples.

| Method | Avg | In distribution | | | Out Of Distribution | | | | | | |
|---|---|---|---|---|---|---|---|---|---|---|---|
| | | MedMCQA | MedQA | PubMedQA | MMLU-Pro | GPQA | Lancet | MedBul4 | MedBul5 | MedXpert | NEJM |
| Random | 55.7% | 58.6% | 68.8% | 76.5% | 64.8% | 46.5% | 59.0% | 57.8% | 50.2% | 16.8% | 58.2% |
| | ±0.3% | ±0.4% | ±0.2% | ±0.0% | ±0.2% | ±1.9% | ±1.2% | ±1.0% | ±1.1% | ±0.4% | ±0.0% |
| Learnability | 55.6% | 58.0% | 69.0% | 73.8% | 64.9% | 48.5% | 61.9% | 55.8% | 48.4% | 16.6% | 59.1% |
| | ±0.1% | ±0.4% | ±0.2% | ±0.6% | ±0.6% | ±0.8% | ±0.2% | ±1.9% | ±1.9% | ±0.0% | ±1.7% |
| Middle Perplexity | 56.4% | 57.9% | 69.7% | 76.8% | 65.2% | 47.8% | 60.4% | 57.8% | 50.5% | 16.0% | **61.8%** |
| | ±0.1% | ±0.2% | ±0.7% | ±0.9% | ±0.3% | ±1.7% | ±1.7% | ±1.0% | ±0.8% | ±0.2% | ±1.7% |
| Embedding Diversity | 56.0% | **58.7%** | 68.8% | 76.4% | **65.3%** | 49.5% | 59.8% | 54.7% | 50.2% | 16.1% | 60.2% |
| | ±0.2% | ±0.2% | ±0.7% | ±1.0% | ±0.3% | ±1.3% | ±1.3% | ±0.8% | ±1.5% | ±0.1% | ±1.0% |
| S2L | 56.3% | 58.3% | 67.6% | **77.0%** | 65.0% | 46.0% | **63.5%** | 56.7% | 51.5% | 16.2% | 61.4% |
| | ±0.3% | ±0.3% | ±0.6% | ±0.2% | ±0.4% | ±0.9% | ±1.1% | ±2.4% | ±1.1% | ±0.7% | ±0.7% |
| Ours | **58.0%** | 58.0% | **71.4%** | 75.3% | 65.0% | **49.6%** | 63.2% | **62.8%** | **55.8%** | **17.9%** | 60.7% |
| | ±0.2% | ±0.3% | ±0.0% | ±0.2% | ±0.8% | ±1.4% | ±0.6% | ±0.2% | ±1.0% | ±0.4% | ±0.8% |

*Table 7.* Performance of finetuning Llama-3.1-8B-Instruct on the m23k dataset when selecting 1000 examples.

| Method | Avg | In distribution | | | Out Of Distribution | | | | | | |
|---|---|---|---|---|---|---|---|---|---|---|---|
| | | MedMCQA | MedQA | PubMedQA | MMLU-Pro | GPQA | Lancet | MedBul4 | MedBul5 | MedXpert | NEJM |
| Random | 54.7% | 58.5% | 70.0% | 63.2% | 61.5% | 46.2% | **61.3%** | 57.8% | 52.3% | **18.3%** | 58.1% |
| | ±1.6% | ±1.5% | ±1.6% | ±12.5% | ±1.3% | ±2.1% | ±1.1% | ±1.9% | ±0.6% | ±0.5% | ±1.1% |
| Learnability | 54.3% | 57.8% | 68.1% | 70.5% | 58.4% | 45.7% | 60.3% | 53.5% | 51.8% | 18.1% | 58.5% |
| | ±0.5% | ±1.2% | ±1.0% | ±1.3% | ±1.7% | ±1.1% | ±2.8% | ±1.3% | ±2.7% | ±0.6% | ±0.7% |
| Middle Perplexity | 55.1% | 57.9% | 68.9% | 70.8% | 60.1% | 47.3% | 59.8% | 55.7% | 54.9% | 17.6% | 58.2% |
| | ±0.9% | ±1.0% | ±0.8% | ±1.7% | ±1.9% | ±1.2% | ±0.6% | ±0.5% | ±4.2% | ±0.8% | ±1.5% |
| Embedding Diversity | 54.2% | 58.2% | 70.1% | 61.2% | 61.6% | 45.0% | 61.0% | 56.7% | 52.4% | 17.6% | 58.4% |
| | ±0.3% | ±0.0% | ±0.7% | ±0.8% | ±0.2% | ±0.1% | ±0.6% | ±2.8% | ±0.5% | ±0.6% | ±0.3% |
| S2L | 55.8% | 58.4% | 71.2% | 75.4% | 61.9% | 46.8% | 59.5% | 55.7% | 51.1% | 17.8% | 60.4% |
| | ±0.2% | ±0.3% | ±0.0% | ±0.6% | ±0.3% | ±0.4% | ±0.5% | ±1.8% | ±0.8% | ±0.1% | ±0.8% |
| Ours | **57.5%** | **59.6%** | **74.2%** | **76.2%** | 62.7% | **47.8%** | 61.0% | 59.4% | **55.4%** | **18.3%** | 60.5% |
| | ±0.7% | ±0.4% | ±0.7% | ±0.3% | ±1.0% | ±2.7% | ±0.4% | ±0.6% | ±1.8% | ±1.1% | ±1.2% |
| m1k | 53.1% | 58.4% | **74.2%** | 32.0% | **63.0%** | 46.5% | 60.7% | **61.5%** | 55.0% | 17.3% | **62.5%** |
| | ±0.1% | ±0.2% | ±0.3% | ±1.1% | ±0.5% | ±0.6% | ±0.0% | ±1.1% | ±0.5% | ±0.4% | ±1.2% |

*Table 8.* Performance of finetuning Qwen2.5-7B-Instruct on the OpenThoughts-Math dataset when selecting 1000 examples.

| Method | Avg. | AMC23 | AIME24 | AIME25 | MATH500 |
|---|---|---|---|---|---|
| Random | 43.12% ± 3.19% | 56.88% ± 5.54% | 20.83% ± 5.00% | 16.67% ± 5.44% | **78.10% ± 0.42%** |
| Learnability | 41.73% ± 1.59% | 53.00% ± 5.50% | 20.67% ± 3.06% | 15.67% ± 2.74% | 77.60% ± 2.26% |
| Middle Perplexity | 39.03% ± 1.42% | 51.07% ± 4.53% | 13.33% ± 3.85% | 16.19% ± 4.05% | 75.50% ± 0.42% |
| Embedding Diversity | 43.50% ± 2.05% | 57.86% ± 5.48% | **22.38% ± 4.99%** | 16.67% ± 2.72% | **77.20% ± 1.13%** |
| S2L | 42.72% ± 0.98% | 51.25% ± 1.44% | 16.67% ± 2.72% | **26.67% ± 0.00%** | 76.30% ± 0.58% |
| Ours | **43.91% ± 2.05%** | **59.17% ± 4.15%** | 21.11% ± 3.73% | 18.15% ± 4.75% | **77.20% ± 1.04%** |

