# OpenReview forum: "Reasoning Quality Emerges Early: Data Curation for Reasoning Models"
_ICML.cc/2026/Conference — ICML 2026 regular_

### Official Review · Reviewer_MCKD · 2026-03-12

**Soundness:** 2
**Presentation:** 3
**Significance:** 3
**Originality:** 3
**Overall Recommendation:** 4
**Confidence:** 4

**Summary:**

This paper proposes a multi-stage data selection framework that prioritizes informative samples before full trace generation to address the prohibitive cost of generating large-scale chain-of-thought (CoT) datasets for large language model (LLM) reasoning training. First, perturbed checkpoints of the pretrained model are constructed to simulate diverse training states, and easy samples are filtered out using the loss of initial reasoning tokens. For remaining candidates, loss trajectories across checkpoints are calculated based on truncated sequences; theoretical analysis validates that these trajectories can serve as a proxy for gradient diversity. Samples are then clustered by trajectories and representative ones are selected to balance learnability and diversity. Experiments demonstrate that this strategy drastically cuts token consumption while maintaining or even boosting LLMs' downstream reasoning performance.

**Compliance With Llm Reviewing Policy:**

Affirmed.

**Final Justification:**

The authors have addressed my concerns, and I will increase my score for this accordingly.

**Key Questions For Authors:**

1. The theoretical justification of some significant hypothesis: could you please further explain the "In particular, by perturbing the pretrained model parameters with controlled noise, one can construct a set of checkpoints that effectively emulate models at different stages of fine-tuning" and "As a result, loss value of examples at highly perturbed checkpoints serves as a reliable proxy for their difficulty in the fine-tuning process".
2. Choice of perturbation strategy: The method generates perturbed checkpoints by adding Gaussian noise to model parameters. How sensitive is the method to the magnitude of the perturbation noise（ie: In the method part, this paper indicates that the highly perturbed checkpoints serve as the difficulty indicator of the data. Could you please clarify whether the larger the perturbation magnitude, the better this steering effect will be, or if there exists an upper bound for the perturbation magnitude?) and the number of checkpoints used?
3. The results in Experiments: I noticed that in Table5-Table 8, the performance of a large number of baselines is inferior to that of the random model. Could you please explain the reason?

**Limitations:**

Most importantly, several core premise assumptions lack theoretical justification. Secondly, some experimental performance of this method is unremarkable(especially table5 - table 8), with only marginal improvements over other comparative approaches.

**Strengths And Weaknesses:**

## Strengths

1. The experiments are extensive and detailed, fully verifying that the proposed method can significantly reduce the token overhead of dataset construction while maintaining or even improving the downstream reasoning performance of the model, which reflects high data efficiency in the training phase of reasoning models.
2. The paper has a reasonable overall structure and rigorous writing. The research motivation and full technical process are elaborated logically, and the accompanying figures can accurately assist in conveying core viewpoints, helping readers quickly grasp the research ideas.
3. It directly addresses the key bottleneck in the reasoning training of large language models—the prohibitive cost of large-scale chain-of-thought data generation. The proposed pre-generation selection strategy of reasoning examples can be directly applied to large model training pipelines, providing a feasible solution for reducing training costs and improving training efficiency.
4. It innovatively establishes a rigorous connection between the loss trajectories of perturbed model checkpoints and the gradient similarity of training samples, and verifies through theoretical analysis that loss trajectories can serve as an effective proxy for gradient diversity, laying a theoretical foundation for trajectory clustering and sample selection. The staged design integrating three core components drastically reduces the number of samples requiring full chain-of-thought generation, achieving substantial token cost savings.


## Weaknesses
1. Some key assumptions lack corresponding theoretical derivation processes, and some even fail to complete basic empirical verification. For example, assertions such as "constructing checkpoints by perturbing pre-trained model parameters with controlled noise to simulate different stages of fine-tuning" and "using sample loss values under highly perturbed checkpoints as proxy indicators of fine-tuning difficulty" have not been fully demonstrated.
2. The sections describing the trajectory-based sample selection process and specific implementation details lack detailed explanations and are not clear enough, which is not conducive to the reproduction and expansion of the method by other researchers.
3. No analysis has been conducted on the impact of core hyperparameters such as the number of perturbed checkpoints, token length for difficulty estimation, clustering strategies and perturbation upper bound on model performance. The influence mechanism and optimal value range of hyperparameters have not been clarified, which limits the practical application of the method.

---

> ### Author Rebuttal · Authors · 2026-03-31
>
> We would like to thank the reviewer for their feedback and appreciate the acknowledgement of our experiment design, overall structure and innovation. We're happy to clarify questions raised by the reviewer.
>
> 1. On the justification on the perturbations
>
> 	- Why do perturbed checkpoints emulate different stages of finetuning? The key idea is that the loss is smooth and flat (relatively linear) near the pretrained checkpoints, so training dynamics (loss landscape and gradients) are similar during finetuning and within the area around the pretrained model containing the perturbed checkpoints. We justify this with an additional empirical evaluation: We randomly perturbed 4 checkpoints and also sampled 4 checkpoints during the actual finetuning of the model and evaluate loss on them. We then fit a quadratic function between the x, the norm of checkpoint displacement (from pretrained model), and y, the loss of the samples on the checkpoint. The result gives a coefficient of the second order term as small as $1.15\times 10^{-3}$ and a coefficient of determination $R^2=0.9377$. This indicates that 1) the loss curve is flat as the quadratic term is negligible and 2) the finetuned and perturbed checkpoints all fall in the same smooth region. Given the observation of the landscape, theoretically the loss behavior of individual examples on perturbed checkpoints will be similar to the loss behavior during finetuning, except being scaled.
> 	- Why does loss value of examples at highly perturbed checkpoints serve as a reliable proxy for their difficulty in the fine-tuning process? By making the model perform worse in that flat quadratic region, we increased the loss and made the easy and difficult problems more separable. The correlation between loss and difficulty is also empirically justified in Fig. 4 of our manuscript.
> 2. On the selection of perturbation strategy
>
> 	We refer the reviewer to our response point 1 to reviewer 5An7. Empirically, we find the average perturbed loss being 2.75x ~4.5x the original loss is a good region with stable correlation. As we further increase the perturbation to >4.5x, the correlation starts to decrease, as we’re moving so far away from the model that loss becomes less informative.
> 3. On the experiments result in table 5-8
> 	- Comparison with Random: The baseline methods we include are principled methods for data selection for instruction tuning data selection, which work well on only instruction tuning datasets with shorter responses but fail in longer reasoning (see line 62-68) and were outperformed by random selection.
> 	- Comparison with Ours: due to the large variability in size and type of benchmarks, there is not a single baseline that performs best on all of them, and average performance is considered a more reliable metric. Furthermore, as discussed in point 4 to reviewer KsjF, our method significantly reduces the cost of generating SFT data. On m23k, we can reduce 17.5 H200 GPU hours while improving the reasoning performance by an average of 2%. We believe this is indeed very significant and cannot be achieved by any of the baselines that can only select data from a large pool of reasoning traces.
> 4. Detailed Implementation
>
> 	We first evaluate the loss of the data on the $n$ perturbed checkpoints we get in Section 3.1 as in Eq.6. Then within each data source, we cluster with FAISS the loss based on these $n$ loss values, where the number of clusters is the number of data to select (given by Eq. 5) divided by the hyperparameter $k$, which we gave ablation results in Figure 8. Finally, within each cluster, we sample the data based on Eq. 8, and take $k$ samples from each cluster with the highest difference in loss.
>
> 	In appendix D, Algs 1–2 details how we handle sources where the budget exceeds available data: we select all items from that source and redistribute the remainder proportional to the budget of other sources. We hope the pseudocode in the appendix would provide enough clarification on the implementation.
> 5. Additional ablation
>
> 	We did have ablation on the clustering hyperparameter along with other ablations in the appendix, see Fig. 8. We include additional ablation results on the number of tokens used for difficulty filtering in our response point 1.b to reviewer 5An7, which implies that the performance is not susceptible to the choice of the perturbation scale as long as it stays in a reasonable region. We also ablate the number of checkpoints used in diversity clustering here. There is some slight drop in performance, but the results remain competitive.
> |# Ckpts|Avg|ID|OOD|
> |:-|:-|:-|:-|
> |8|57.77%|69.26%|52.85%|
> |6|56.98%|68.04%|52.24%|
> |4|56.57%|67.77%|51.77%|
>
> We hope that our clarifications have addressed all the concerns and demonstrated the soundness of our method, and we hope that in light of our clarifications the reviewer considers increasing their support for our work. Please let us know if we can provide any further clarification.

---

> > ### Author Rebuttal · Reviewer_MCKD · 2026-04-03
> >
> > Thank you for the authors' detailed rebuttal response.
> >
> > The authors explained that the baselines performed worse than random selection because these methods only work well on instruction tuning datasets with shorter responses. This is perplexing. Why do these methods only apply to instruction tuning datasets with shorter responses, and why didn't the authors choose baselines that are effective on instruction tuning datasets with longer responses? Are there really no relevant baselines available?

---

> > > ### Author Response · Authors · 2026-04-07
> > >
> > > We thank the reviewer for the question. We further clarify below how data selection in reasoning data is fundamentally different from instruction tuning data.
> > >
> > > **Instruction tuning vs reasoning data**
> > >
> > > Instruction-tuning data consists of (instruction, output) pairs and improves a model's ability to follow human instructions, generalize to new tasks, and respond naturally to prompts. For some tasks, inserting “let’s think step by step” in the prompt urges the model to *think for a few steps* before outputting the answer, but the entire sequence is generally shorter than 1500 chars.
> > >
> > > On the other hand, reasoning models are trained to produce very long chain-of-thoughts *by default* (without having to insert “let’s think step by step” in the prompt) with up to 50-100k chars, to solve *highly complex* tasks beyond following instructions.
> > >
> > >
> > > Below, we provide the average length of instruction tuning and reasoning datasets use in recent work (IT for instruction tuning):
> > > |Paper|Dataset|Dataset Type|Avg. Response length (char)|
> > > |---|---|---|---|
> > > |Liu et al., 2024|Deita (Selected from mixed data pool)|IT|1242|
> > > |Chen et al., 2024b|ALPAGASUS (Selected from Alpaca)|IT|339|
> > > |Zhou et al., 2025|DavIR (Selected from Alpaca)|IT|270|
> > > |Liu et al., 2025|Selected from CMedQA, LawQA & FinQA|IT|106, 234, 94|
> > > |Bhatt et al., 2024|Selected from FLAN V2|IT|139|
> > > |Zhang et al., 2025|Selected from a mixture; statistics not given|IT|/|
> > > |Yang et al., 2024|Selected from MathInstruct|IT|435|
> > > |Huang et al., 2025|m23k|**Reasoning**|**6.2k**|
> > > |Guha et al., 2025|OpenThoughts-math|**Reasoning**|**20k**|
> > >
> > > Unlike instruction tuning data, reasoning trajectories are highly exploratory and noisy: Throughout the long chain-of-thought, the model frequently makes multiple failed attempts, pauses to re-evaluate (e.g., generating “Wait, but the problem says…” or “Hmm, let me double check…”), and tries multiple ways to approach the problem before arriving at a solution. As pointed out in Line 60-69, this inherent noise disrupts traditional data selection metrics: When loss or perplexity is averaged over a long reasoning trajectory, the abundance of easy-to-predict tokens dilutes the high-perplexity tokens where the model is genuinely uncertain, making overall loss/perplexity a poor indicator of difficulty. Similarly, the presence of multiple different solution attempts within a single response makes the embedding of the whole trajectory noisy and unrepresentative. Because standard instruction-tuning data is inherently shorter and cleaner, existing baselines simply are not applicable to long noisy reasoning data, which is exactly what our empirical results confirm in table 1-3, where random baseline outperforms some instruction-tuning baselines in some settings. We note that there is no separate baselines for selecting short and long instruction tuning data as instruction tuning data is generally limited in length.
> > >
> > > **Data selection for reasoning data**
> > >
> > > As discussed in the second paragraph of our Introduction (L28-42): “reasoning traces are either manually curated or generated by strong reasoning models, and subsequently filtered to promote diversity and difficulty. Diversity is often enforced by categorizing data using more capable LLMs. Difficulty-based filtering either prompts an LLM to estimate the difficulty of each question and retains only the hardest instances, or selects questions that elicit the longest model-generated reasoning traces. However, producing large volumes of long chains of thought followed by extensive LLM-based filtering, renders the construction of high-quality SFT datasets prohibitively expensive at scale.”
> > >
> > > Guha et al. (2025) concluded that LLM-filtering is more effective than picking the longest responses, and our Figure 6 and Table 2 already compare with m1K, which is an LLM-filtering baseline for difficulty and diversity. Here we also include the “longest reasoning” baseline to our Figure 6. We follow the same setting, select 250, 500, 750 and 1000 examples from the M23k data to finetune a Qwen2.5-7B-Instruct model, and present the average performance across 10 different medical benchmarks in the following table (LR for Longest Reasoning). We see that taking the longest response is not a good metric for reasoning data selection.
> > >
> > > |Selection Budget|Method|Avg|ID|OOD|
> > > |:---|:---|:---|:---|:---|
> > > |250|Ours|51.78%|65.21%|46.02%|
> > > |250|LR|41.56%|51.79%|37.18%|
> > > |500|Ours|55.67%|67.54%|50.59%|
> > > |500|LR|49.66%|61.87%|44.42%|
> > > |750|Ours|56.52%|68.50%|51.38%|
> > > |750|LR|53.60%|65.61%|48.45%|
> > > |1000|Ours|57.77%|69.26%|52.85%|
> > > |1000|LR|54.58%|67.13%|49.20%|
> > >
> > > Notably, methods for selecting reasoning data (Huang et al.,2025; Guha et al.,2025), whether based on response lengths or LLMs, require the full instantiation of long reasoning response. But, our method only uses the initial part of the reasoning traces to filter difficult and diverse examples. Thus our method is up to 10 times faster than these baselines, as we showed in Fig. 7.

---

### Official Review · Reviewer_ox2j · 2026-03-12

**Soundness:** 2
**Presentation:** 1
**Significance:** 2
**Originality:** 2
**Overall Recommendation:** 3
**Confidence:** 5

**Summary:**

This paper study the data selection for training reasoning models. It uses the difficulty and diversity of reasoning traces can be identified early during generation, without generating full chain-of-thought sequences. The method uses perturbed pretrained checkpoints as proxies for fine-tuning trajectories, filters difficult examples using loss on the first 100 tokens, and clusters examples by loss trajectories across 1k tokens for diversity. Experiments on medical and math benchmarks show up to 2% improvement over baselines while generating only 9% of tokens.

**Compliance With Llm Reviewing Policy:**

Affirmed.

**Final Justification:**

I have increased my score to 3. However, the presentation of the paper would benefit from further improvement. For example, the results could be accompanied by clearer descriptions and explicit takeaways, rather than presenting tables without sufficient interpretation or discussion. I share similar concerns to those raised by **Reviewer 5An7** regarding the clarity of the presentation.

**Key Questions For Authors:**

1. Lines 10–12 claim that RL requires "very large data" while SFT uses a "smaller set" of reasoning traces, but no reference is provided to support this distinction. I am not sure whether this is correct claim.
2. The variable `e` appears in Equation 1 and again in line 638 of the Appendix but is not being defined.
3. The clustering method used to partition examples by loss trajectory is not specified.

**Limitations:**

yes

**Strengths And Weaknesses:**

Strengths:
* Soundness: The core idea of using perturbed checkpoints as a proxy for fine-tuned models is clever. This resolves the need for reference-model-based methods in data selection and measuring data attributes, as seen in techniques such as [1][2][3]. I will encourage authors to discuss them in the related work, especially how we can erase concerns about dependency on reproducing fine-tuned models.
* Presentation:
* Significance: Data curation is essential to the field for saving budgets and better allocating training resources.
* Originality: I appreciate the idea of using perturbed checkpoints as a proxy.

Weaknesses:
1. While I think the idea of using perturbed checkpoints as a proxy is novel and clever as an alternative to actually fine-tuning the model, the use of learnability/difficulty as a selection criterion has already been explored in many data selection works [1][3].
2. Section 2 (lines 60–68) argues that existing instruction-tuning data selection methods fail on reasoning data due to trace length noise, but does not empirically demonstrate this failure. I think there is a need to incorporate these methods in the experiments. These methods have shown improvements on their own benchmarks, and the argument that long CoTs introduce noise does not directly answer why they underperform. Additionally, the last paragraph of Section 2 is cut off.
3. Math notation is introduced incrementally and inconsistently, making the method section hard to follow. I suggest having a clear roadmap or a formal problem setup at the beginning of Section 3.
4. Repetitive statements (e.g., lines 302–312 largely restate earlier content) reduce readability.
5. Figure 5 shows a drop in Pearson correlation at 0.5k tokens, which makes the choice of using only the first 100 tokens for difficulty estimation in Figure 4 confusing and insufficiently justified.
6. Figure 7 is difficult to interpret and should be redesigned. The accompanying discussion is also insufficient.
7. Tables 1, 2, and Figure 6 are presented without any accompanying result discussion in the text. Results should not be placed in the paper without analysis.
8. I think the structure of Appendix should be improved. Algorithm 1 and 2 are disconnected to the main paper.

[1] https://proceedings.mlr.press/v162/mindermann22a/mindermann22a.pdf
[2] https://arxiv.org/pdf/2501.06708
[3] https://arxiv.org/abs/2404.07965

---

> ### Author Rebuttal · Authors · 2026-03-31
>
> We would like to thank the reviewer for their feedback and acknowledging the novelty of perturbation. We believe important ideas and details behind our method may have been misunderstood and we're happy to clarify below.
>
> 1. On the use of the learnability score and failure of instruction tuning data selection method.
>
>     Contrary to the reviewer’s comment, we extensively included data selection methods on instruction tuning datasets in our experiments, which include [learnability](https://arxiv.org/abs/2310.13008), [middle perplexity](https://arxiv.org/abs/2309.04564), and [embedding](https://arxiv.org/abs/2401.06692). The empirical results demonstrate that these methods won’t work for reasoning data. Specifically, our selection **outperforms learnability baselines by 6% on medical dataset with Qwen2.5-7B-Instruct (Table 1), 3% on medical dataset in Llama-3.1-8B-Instruct (Table 2), and 5% on the math dataset with Qwen (Table 3)**. The learnability baseline was even outperformed by random baseline in Table 1.
>
>     Thus for reasoning data, **we proposed a new definition of learnability** in Eq. 4, which differs from the definition in literature (see lines 357-361) and depends on the difference in loss of truncated reasoning in least and most perturbed checkpoints, without finetuning the model.
>
> 2. On the interpretation of Fig. 4 and Fig. 5.
>
>     The reviewer asks why we should use 100 tokens for difficulty sampling when even 500 tokens doesn’t correlate well with the full loss. We want to emphasize that difficulty filtering and diversity sampling are separate phases and serve different purposes.  As pointed out in L210-218, the 100-token loss should correlate with the problem difficulty, not the loss on the full response; On the other hand, as analyzed in line 241-244, the loss for diversity sampling has to correlate with full loss to identify data that has a similar effect on the training dynamics (gradients).
>
> 3. Interpretation of Figure 7
>
>    We demonstrate that full teacher responses are unnecessary for selection, significantly saving costs. Each bar in both subplots in Fig. 7 corresponds to a generation step in Fig. 1. The blue blocks represent the reasoning tokens generated by the teacher up to this phase; the red ones indicate pruned tokens via "early termination" in Fig. 1, and the green blocks show the number of tokens in the final selected SFT data. Our pipeline saved 25.2M (Medical) and 322.6M (Math) teacher tokens, which is equivalent to 21h and 245h on H200 GPU hours, demonstrating its efficiency. We will redesign Fig 7 in our revision.
>
> 4. Comparison of RL and SFT data requirement
>
>     There is recent ample evidence in the literature, as we have already cited in the first and second paragraph of our introduction as well as section 2.2 of our related work. These papers confirm that SFT data should be small and high-quality, i.e. hard and diverse (Muennighoff et al., 2025; Guha et al., 2025; Huang et al., 2025b, Akter et al., 2025); and RL data should be large (OpenAI o1, DeepSeek R1). All these references are already included in our paper.
>
> 5. Clarification on the writing and details
>
>     We have a roadmap at the beginning of Section 3 with Figure 1 which is found helpful by reviewer KsjF. While we didn’t find any inconsistency in our notations, we will proofread again for our revision.
>
>     Specific clarifications include:
>     - Thanks for pointing out the typo in Section 2, A “We” should be added at the beginning of the last sentence.
>     - $e$ in Eq. 1: It's a normalized ones vector (normalized to be norm 1) to ensure that the perturbation is balanced across different layers and that the perturbation remains a reasonable size.
>     - Appx D: Alg. 2 explains how diversity clustering is implemented, and Alg. 1 details how we handle sources where the budget exceeds available data: we select all items from that source and redistribute the remainder proportional to softmax scores or equally across clusters. We’ll add this explanation in the appendix.
>     - Clustering: We use FAISS to cluster losses. The number of clusters per source is $(\text{samples to select}) / 4$ (see L305-307)
>     - Tables 1, 2, and Figure 6. We apologize for not including references in the main body. Fig. 6 demonstrates our method selects better examples compared to other baselines most of the time even when we change the number of data to select. Tables 1-3 confirms the effectiveness of our method when applied to different model architecture and datasets.
>
> We hope that our clarifications have addressed all the concerns and demonstrated that our method is principled and theoretically-motivated, and we hope that in light of our clarifications the reviewer considers increasing their support for our work. Please let us know if we can provide any further clarification.

---

> > ### Author Rebuttal · Reviewer_ox2j · 2026-04-04
> >
> > Thank you for the clarification and for pointing out the baseline papers. Upon reviewing Section 4.2, I noticed that the corresponding citations for each baseline method appear to be missing. I would kindly suggest that the authors consider revising and reorganizing this section, including a clearer presentation of the experimental setup and results, to improve overall clarity.
> >
> > Additionally, the Appendix currently lacks sufficient explanatory context---it would be helpful if the authors could rewrite and provide accompanying descriptions. Overall, I believe the manuscript would benefit significantly from a thorough revision of the writing. Thanks.

---

> > > ### Author Response · Authors · 2026-04-07
> > >
> > > We thank the reviewer for the suggestion. However, we believe that most of the reviewer’s initial and follow up concerns may have stem from overlooking discussions that are already included in our original manuscript (as also recognized by reviewer KsjF). We acknowledge that the reference to Fig. 6 was originally missed, and we thank the reviewer for pointing that out. But, this is a minor oversight and doesn’t necessitate a thorough revision of the manuscript.
> > >
> > > - Regarding baselines, experiments, and appendix:
> > >
> > >     The baseline methods in our experiments are representative of the methods described and cited in section 2 (related work), and we provided *additional citations* during the rebuttal, for popular methods including learnability (Zhou et al.,2025), loss / perplexity (Li et al., 2024b) and embedding based method (Liu  et al., 2024; Bhatt et al., 2024). Our original submission also includes details on the implementation of these baselines. See L355-357 for leanability, L365-367 for perplexity, and L373-377 for embedding.
> > >
> > >     Our presentation of the experimental setup is thoroughly detailed in section 4.1 and Appendix B, with sufficient details such as the dataset we use for selection and benchmarking, and the hyperparameters we used. Our method has been substantially described, both in a high level as in Figure 1 and in detail as in Section 3. We believe the experimentation section already provides sufficient details for reproducibility and offers strong evidence of the effectiveness of our method.
> > >
> > >     In the appendix, we’ve provided sufficient details on the proof of our theorem and explanation on experiment setup details that could not fit in the main paper due to space constraint, including citations to benchmarks and the hyperparameters used. Table 5-8 provide the raw data for Figure 6 and Table 1; Figure 8 provides ablation study with analysis in the caption on what the results imply; and Section D provides the pseudocode to further improve the reproducibility in section 3.3.
> > >
> > > We hope these clarifications provide a more complete picture of our work’s novelty and significance, and we kindly invite the reviewer to re-evaluate the manuscript in light of these points.

---

### Official Review · Reviewer_KsjF · 2026-03-13

**Soundness:** 3
**Presentation:** 3
**Significance:** 3
**Originality:** 3
**Overall Recommendation:** 5
**Confidence:** 4

**Summary:**

The paper proposes an efficient pipeline for curating  SFT data for reasoning models by identifying high-quality samples early in the generation process. Rather than generating full, expensive CoT and then filtering, the authors introduce a three-stage "early-exit" curation strategy:
- Difficulty Filtering: The model generates only the first 100 tokens of a reasoning trace. Difficulty is estimated by measuring the loss of these tokens on a "highly perturbed checkpoint" of the pretrained model. High loss under large parameter noise indicates the example is challenging and requires precise adjustments.
- Diversity Sampling: For the difficult traces, the model generates up to 1k tokens. The authors prove that examples with similar loss trajectories across noisy checkpoints induce similar gradients during fine-tuning. They cluster these trajectories and sample from each to ensure gradient diversity.
- Final SFT Generation: Full reasoning traces are only generated for this final, optimized subset

**Compliance With Llm Reviewing Policy:**

Affirmed.

**Final Justification:**

Authors responded to my concerns.

**Key Questions For Authors:**

The paper assumes that difficulty can be reliably identified within the first 100 tokens. How does this method handle complex reasoning tasks where the "bottleneck" or the most difficult logical leap occurs much later in the trace (e.g., in the middle of a long mathematical proof or a complex differential diagnosis)?

The experiments focus on 7B and 8B models. As model size increases the loss landscape often becomes sharper. Have the authors tested if the same perturbation scaling factors and the 1k-token diversity cutoff hold for significantly larger architectures?

**Limitations:**

Yes

**Strengths And Weaknesses:**

Soundness
- Strength (Theoretical Grounding): The paper is not just heuristic-driven; it provides a formal proof (Theorem 3.1) connecting loss trajectories at perturbed checkpoints to upper-bounded gradient differences. This provides a solid mathematical basis for using loss as a proxy for gradient similarity.
- Strength (Empirical Breadth): The authors validate the method across two very different domains: math and healthcare which suggests the "early signal" in CoT is a generalizable property.
- Weakness (Sensitivity to Assumptions): The method relies heavily on the "locally smooth and nearly flat" loss landscape assumption around the pretrained initialization. While common in SFT literature, this may not hold for highly specialized or shifted domains where the initialization is far from a low-loss basin.

Presentation
- Strength (Clarity of Pipeline): Figure 1 and Algorithm 2 provide a very clear visual and logical roadmap of the curation process. The explanation of "perturbed checkpoints" as level sets of the loss landscape is intuitive.
- Weakness (Evaluation Redundancy): Similar to many current papers, the results section repeats much of the data from the tables (Tables 1-3) in prose, which can be dense to read.

Significance
- Strength (Economic Impact): The reduction in token generation is massive—91.4% fewer tokens for the math dataset. For teams training large-scale reasoning models, this represents a significant reduction in API costs or GPU hours.
- Weakness (Marginal Accuracy Gains): While the efficiency is impressive, the absolute accuracy gains are relatively modest. The primary value proposition here is cost-efficiency rather than a paradigm shift in final model performance.

Originality
- Strength (Novel Proxy): Using Gaussian noise perturbations of the current model to simulate fine-tuning trajectories, without actually performing a training run—is a clever and original way to bypass the need for expensive proxy models or full SFT.
- Strength (Creative CoT Use): Recognizing that the first 100 tokens of a Coated response act as a "strategic summary" that anchors the difficulty of the entire problem is an insightful observation of how reasoning models operate.

---

> ### Author Rebuttal · Authors · 2026-03-31
>
> We would like to thank the reviewer for the detailed review and support of our work. We’re happy to clarify the questions raised by the reviewer:
>
> Q1. As pointed out in Section 3.2 (line 218, 219, 184, 185) , we aim to remove **questions** that are simple for the student model in the difficulty filtering phase. Since the model response usually begins by summarizing the key anchor points for reasoning, if even a perturbed (deteriorated) student model achieves a low loss, then this problem must be easy and straightforward, and it is unlikely that the reasoning will lead to a convoluted bottleneck later on. Therefore, filtering by the difficulty of the question through the loss of the first 100 tokens remains an effective and efficient method.
>
> Q2. Good question! In the literature, there is ample support that the loss landscape actually becomes more smooth as the model size increases. This has been empirically verified by Chen et al., 2025a, in which the authors empirically compared the “basin size” of the loss of LLMs varying in size from 0.5B to 32B. Their findings indicate that larger LLMs tend to have larger basins, meaning that the smooth assumption would still apply for larger models. Empirically due to computational constraints, we’re not able to conduct experiments on larger models, but we would expect similar results given the assumptions on the loss landscape being verified.
>
> Also we would like to explain the weaknesses pointed out by the reviewer:
>
> 3. Sensitivity to assumption
>
> 	Even for highly specialized domains, for modern pretrained LLMs with enough coverage in their pretraining data, the relative change in fine-tuning is much smaller than pretraining. In fact as empirically verified in Chen et al., 2025a,  [Brutzkus et al.](https://arxiv.org/abs/1810.03037), [Kiselev et al.](https://arxiv.org/abs/2409.11995) and [Li et al.](https://arxiv.org/abs/1712.09913), most SFT domains are like “sub-basins” within the large basin of the loss landscape of the pretrained model. Thus, our assumptions will remain valid even for specialised domains.
> 4. Marginal accuracy gains
>
>     All baseline approaches require the presence of full reasoning trajectories, which in practice takes a very long time to generate (much larger than the selection cost). Take the Openthoughs-math dataset as an example. The teacher examples were generated by DeepSeek-R1, which would take around 245 H200 GPU hours to fully generate, while our method reduces this to around 21 GPU hours (91% reduction). With only 9% of the cost and the time, the fact that our method achieves 2% higher performance on medical reasoning and matches the math reasoning is indeed significant. Nevertheless, we agree that the primary value is cost-efficiency.

---

> > ### Author Rebuttal · Reviewer_KsjF · 2026-04-02
> >
> > They responded to my second question with a citation I was not aware of. Thank you for that

---

### Official Review · Reviewer_5An7 · 2026-03-13

**Soundness:** 1
**Presentation:** 2
**Significance:** 3
**Originality:** 3
**Overall Recommendation:** 4
**Confidence:** 2

**Summary:**

The paper propose a new data selection pipeline for supervised fine tuning which selects a small and high quality data points. At high level, the pipeline consists of two steps: (1) difficulty filtering (2) diversity sampling. For these two steps they propose to use perturbed version of the pretrained model. Starting from initial weights $\theta_0$ of the pretrained model, the authors iteratively perturb model parameters by using Gaussian noise and obtain $\theta_1, \dots \theta_n$. They select the magnitude of the noise in a way that first and the last checkpoints are substantially different.

The key observations are as follows. First, the log likelihood of the first 100 tokens for the most perturbed model aligns well with the difficulty of the problem. They suggest using this as a proxy for measuring the difficulty of the problem in the difficulty filtering step. Second, the authors compute a log-likelihood vector per prompt (up to 1000 tokens) across all perturbed checkpoints and argue (and formally prove) that two examples with similar such vectors will have similar gradients during fine-tuning. They compute these vectors and use as a measure of similarity then apply diversity sampling based on these similarities.

Experiments are conducted on medical and math reasoning benchmarks. They compare their method against various other data sampling techniques: random, Learnability, Middle Perplexity, Embedding Clustering, M1k. The proposed method achieves performance close to full-data training on medical benchmarks and in this way they use significantly fewer tokens across all pipeline stages. For math reasoning, their results are comparable to embedding clustering method with the same sample size.

**Compliance With Llm Reviewing Policy:**

Affirmed.

**Final Justification:**

Updated after rebuttal. My main concern was statistical soundness, which the authors addressed. That said, for a paper of this type, statistical robustness is essential, and the paper still needs significant rewriting. I updated my score to weak-accept given the interesting contribution.

**Key Questions For Authors:**

- Figure 4 looks noisy and its conclusions feel arbitrary. Why is the difference between "First 100 tokens, Most Perturbed Checkpoint" and "Full Response, Most Perturbed Checkpoint" at difficulty level 5? Have the authors sampled 10–20 perturbed models (or more) to verify that the most perturbed model with first 100 tokens consistently aligns with true difficulty Why was 100 tokens chosen specifically? Is this threshold principled? Do results remain stable if it is varied to 101, 102, or a random value in the 100–120 range?
- Table 1 reports results from a single training run. Can authors provide confidence intervals for their results?
- How do the FLOPs of the proposed pipeline compare against embedding-based methods? Given that diversity sampling requires multiple log-likelihood computations across all perturbed checkpoints, is the method truly more efficient?
- Does an analogue of Theorem 3.1 hold for embedding similarity? More formally, does embedding similarity between two examples imply anything about their gradient alignment?

**Limitations:**

yes

**Strengths And Weaknesses:**

Strengths:
- The pipeline is novel and practically motivated, offering a principled approach to SFT data selection.
- Experiments show that this pipeline cuts token counts significantly for the training and decision phases.

Weaknesses:
- The computational cost of the pipeline is not discussed or compared against baselines such as embedding-based or perplexity-based sampling. Diversity sampling step requires multiple log-likelihood computations across all perturbed checkpoints. A FLOPs comparison would be important to fairly assess the method's practical competitiveness.
- The experimental results (tables, figures) lack statistical rigor. No confidence intervals or statistical tests are reported. This makes it difficult to assess whether the observed performance differences are reliable or simply artifacts of a single run.

---

> ### Author Rebuttal · Authors · 2026-03-31
>
> We would like to thank the reviewer for their review and acknowledging our novel framework. We believe important ideas behind our method may have been misunderstood and we're happy to clarify below.
> 1. Using perturbed checkpoint as difficulty filtering
>
>    a. Why is loss on the first 100 tokens a better indication of the difficulty?
>
>    - As presented in Fig. 2 and discussed in line 213-218, the first 100 tokens is almost always a rephrase of the problem, grounding the reasoning. On the other hand, using the full response is not only inefficient, but also ineffective, as tokens associated with complex reasoning steps, which leads to a high loss, can be diluted by the trivial tokens with low loss.
>
>    b. Additional experiment to verify the stability of this correlation
>
>    - We include additional experiments here to demonstrate that 100~120 tokens provide a very reliable estimate of difficulty when the loss of perturbed checkpoints remain within a factor of 2.75x - 4.5x the original loss. Specifically, we perturbed the checkpoints independently at 15 different scales, and evaluated the loss on the first 100, 110, and 120 tokens. We then calculate the average loss grouped by the LLM-annotated difficulty, and calculate the correlation of the mean loss with the difficulty level. Since we measure the average loss of the first 100 tokens, not just the 100th token, using 101 or 102 tokens would have minimal impact.
>
> Table 1. Correlation between loss of 100 tokens and difficulty
> |loss increment|correlation|
> |:------|-----:|
> |1.92x|0.73|
> |2.04x|0.44|
> |2.23x|0.76|
> |2.30x|0.3|
> |2.49x|0.72|
> |2.64x|0.94|
> |2.73x|0.97|
> |3.01x|0.94|
> |3.30x|0.9|
> |3.42x|0.85|
> |3.53x|0.92|
> |3.84x|0.93|
> |4.32x|0.83|
> |4.68x|0.78|
> |4.94x|0.53|
>
>
> Table 2. Correlation between loss of 110 tokens and difficulty
> |loss increment|correlation|
> |:-------|------:|
> |2.01x|0.66|
> |2.13x|0.33|
> |2.34x|0.68|
> |2.39x|0.07|
> |2.60x|0.68|
> |2.74x|0.88|
> |2.85x|0.97|
> |3.16x|0.92|
> |3.46x|0.87|
> |3.59x|0.81|
> |3.71x|0.91|
> |4.02x|0.93|
> |4.52x|0.79|
> |4.90x|0.68|
> |5.15x|0.44|
>
> Table 3. Correlation between loss of 120 tokens and difficulty
> |loss increment|correlation|
> |:----------|-----:|
> |2.10x|0.58|
> |2.22x|0.22|
> |2.45x|0.59|
> |2.47x|-0.12|
> |2.71x|0.63|
> |2.84x|0.8|
> |2.97x|0.96|
> |3.30x|0.9|
> |3.62x|0.83|
> |3.75x|0.77|
> |3.89x|0.9|
> |4.18x|0.92|
> |4.70x|0.74|
> |5.11x|0.55|
> |5.34x|0.34|
>
>    c. The choice of 100 tokens
>
>    * We did an ablation on the number of tokens used for difficulty filtering, and empirically, 100 tokens are the best:
> | Configuration | Avg | ID (3 benchmarks) | OOD (7 benchmarks) |
> | :--- | :--- | :--- | :--- |
> | 50 tokens | 56.12% | 67.25% | 51.36% |
> | 100 tokens | 57.77% | 69.26% | 52.85% |
> | 200 token | 56.02% | 67.75% | 50.99% |
>
> 2. Confidence intervals:
>
>    We present here the std across three runs of our model on the m23k data:
>    | Avg | ID (3 benchmarks) | OOD (7 benchmarks) |
>    | :--- | :--- | :--- |
>    | 0.47% | 0.25% | 0.56% |
>
>    Since there are multiple benchmarks for the dataset and the results are aggregated over them, the performance reported in the paper is accurate and stable. We'll add stds to our tables in our revision.
>
> 3. Computational costs of the proposed method
>
>    If we consider the setting of the openthoughts-math dataset and exclude the cost of teacher example generation, embedding based method requires $5.0\times 10^{18}$ FLOPs while our method requires $3.0\times 10^{18}$, which is already better. If we further include the cost of teacher generation using DeepSeek-R1, the embedding method would require $3.1\times 10^{19}$ FLOPs whereas our method requires  $5.2\times 10^{18}$ FLOPs. As shown in Fig. 7, our method saves 91% of the generation cost which dominates embedding-based selection.
>
> 4. Possibility of using embedding for gradient alignment
>
>    There are three reasons that embeddings do not provide an accurate estimate of the gradient similarity.
>
>    a. The embedding layer disregards the final LM head layer, which is an important component during SFT. It takes up a non-negligible part of the parameters, and is heavily updated during SFT.
>
>    b. The embedding of the pretrained checkpoint only captures the similarity at the beginning of SFT and does not capture how similarities evolve during SFT.
>
>    c. In practice embeddings are high-dimensional vectors, which makes meaningful clustering inaccurate due to the curse of dimensionality. Even if they intrinsically fall in a low-dimensional manifold, it is not trivial to identify that manifold and project the embeddings onto it.
>
>    Our experiments in Table 1, 2 and Fig. 6 confirm that our approach significantly outperforms embedding-based methods.
>
> We hope that our clarifications have addressed all the concerns and demonstrated that our method is principled and theoretically-motivated, and we hope that in light of our clarifications the reviewer considers increasing their support for our work. Please let us know if we can provide any further clarification.

---

> > ### Author Rebuttal · Reviewer_5An7 · 2026-04-04
> >
> > Thanks for the response. Most of my questions are addressed, and I updated my score to weak-accept. The contribution is interesting. My main worry was about statistical consistency, and the response addressed that. However, I still think the paper would benefit from a significant rewriting effort to make the conclusions statistically sound.

---

> > > ### Author Response · Authors · 2026-04-07
> > >
> > > We thank the reviewer for reading our response and we are glad that our response could address the concerns. We will make sure to include the correlation results and standard deviations to the paper and improve the clarity of Fig 7 detailing the computational costs of our method vs baselines. We appreciate your feedback and we believe these clarifications strengthen the manuscript’s presentation without altering the fundamental scope or outcomes of our work.

---

### Decision · Program_Chairs · 2026-04-30

**Decision:**

Accept (regular)

**Comment:**

The paper shows that SFT data for reasoning models can be curated efficiently without the need to generate full, expensive CoT sequences. The authors introduce a pipeline that utilizes perturbed checkpoints of a pretrained model to estimate the difficulty of an example using only its first 100 tokens, and then clusters examples by their loss trajectories across 1,000 tokens to ensure data diversity.

## Comments

On the positive side, the reviewers found that the proposed method is backed by strong theoretical grounding, specifically Theorem 3.1, which connects loss trajectories to gradient differences. The reviewers liked the breadth of the evaluations across mathematical and medical domains, and they highlighted the token efficiency of the approach, which reduces generation costs by up to 91% while maintaining or slightly improving downstream reasoning performance.

I personally feel like this paper should include (i) results on larger open source models, and newer models, such as Qwen 3 or GPT-OSS or Gemma 3. This would strengthen the claims. I also think the paper needs some more ablations and/or comparisons to other data pruning methods.

The reviewers identified areas of improvement, such as the overall presentation and clarity of the manuscript. Specifically, the discussion led to suggestions:

- Standardize the math notation in the methodology section (and better connect the main paper to the appendix). Integrate the pseudocode and algorithms from the appendix with the main text for clarity.

- Redesign visualizations, e.g., Figure 7, for better interpretability, and ensure all tables and figures are accompanied by some analysis in  main text.

- Incorporate the citations and discussion of the baseline methods (such as learnability and embedding-based techniques) into the main related work and experiment sections.

- There are some typos, e.g., "OpehThoughts"

These are a few examples, but in general, the authors should read the whole set of reviewer comments and revise the paper for the final version.

## Recommendation

Based on these reviews, I recommend accepting this paper. The reviewers are excited about the originality of the framework, noting that utilizing noise perturbations of the current model to simulate fine-tuning trajectories is a creative way to bypass the need for full SFT or proxy models. They like the practical / efficiency impact that this pipeline gives.